# ANGPTL4 mediates shuttling of lipid fuel to brown adipose tissue during sustained cold exposure

Wieneke Dijk[1], Markus Heine[2], Laurent Vergnes[3,4], Mariëtte R Boon[5,6], Gert Schaart[7], Matthijs KC Hesselink[7], Karen Reue[3,4], Wouter D van Marken Lichtenbelt[8], Gunilla Olivecrona[9], Patrick CN Rensen[5,6], Joerg Heeren[2], Sander Kersten[1]*

[1]Nutrition, Metabolism and Genomics group, Division of Human Nutrition, Wageningen University, Wageningen, The Netherlands; [2]Department of Biochemistry and Molecular Cell Biology, University Medical Center Hamburg-Eppendorf, Hamburg, Germany; [3]Department of Human Genetics, David Geffen School of Medicine, University of California, Los Angeles, Los Angeles, United States; [4]Molecular Biology Institute, University of California, Los Angeles, Los Angeles, United States; [5]Department of Medicine, Division of Endocrinology, Leiden University Medical Center, Leiden, The Netherlands; [6]Einthoven Laboratory for Experimental Vascular Medicine, Leiden Univeristy Medical Center, Leiden, The Netherlands; [7]Department of Human Movement Sciences, Maastricht University, Maastricht, The Netherlands; [8]Department of Human Biology, Maastricht University Medical Centre, Maastricht, The Netherlands; [9]Department of Medical Biosciences/Physiological Chemistry, Umeå University, Umeå, Sweden

*For correspondence: sander.kersten@wur.nl

Competing interests: The authors declare that no competing interests exist.

**Abstract** Brown adipose tissue (BAT) activation via cold exposure is increasingly scrutinized as a potential approach to ameliorate cardio-metabolic risk. Transition to cold temperatures requires changes in the partitioning of energy substrates, re-routing fatty acids to BAT to fuel non-shivering thermogenesis. However, the mechanisms behind the redistribution of energy substrates to BAT remain largely unknown. Angiopoietin-like 4 (ANGPTL4), a protein that inhibits lipoprotein lipase (LPL) activity, is highly expressed in BAT. Here, we demonstrate that ANGPTL4 is part of a shuttling mechanism that directs fatty acids derived from circulating triglyceride-rich lipoproteins to BAT during cold. Specifically, we show that cold markedly down-regulates ANGPTL4 in BAT, likely via activation of AMPK, enhancing LPL activity and uptake of plasma triglyceride-derived fatty acids. In contrast, cold up-regulates ANGPTL4 in WAT, abolishing a cold-induced increase in LPL activity. Together, our data indicate that ANGPTL4 is an important regulator of plasma lipid partitioning during sustained cold.

## Introduction

Adipose tissue can be classified into white adipose tissue (WAT) and brown adipose tissue (BAT). Whereas WAT represents the main energy storage organ in the body, BAT is dedicated to the generation of heat via the burning of lipids. BAT is activated during cold exposure, when additional heat production is needed to maintain core body temperature. Heat production by BAT is stimulated via release of norepinephrine by the sympathetic nervous system, causing activation of β-adrenergic signalling and subsequent uncoupling of ATP production from mitochondrial respiration (*Cannon and*

**eLife digest** The body stores energy in the form of fat molecules. Most of these molecules are stored in white fat cells. Other fat cells, the so-called brown fat cells, consume fats and produce heat to maintain body temperature in cold conditions. The capacity of brown fat cells to consume fats has led researchers to investigate whether brown fat cells might be a key to combat obesity.

When an organism is cold, fat is shuttled to the brown fat cells. An enzyme called lipoprotein lipase is involved in a process that allows these fat molecules to be taken up by brown fat cells. However, it was not clear exactly how this process works.

A protein called Angiopoietin-like 4 (ANGPTL4) inhibits the activity of lipoprotein lipase in white fat cells and is also found at high levels in brown fat cells. Here, Dijk et al. used genetic and biochemical approaches to study the role of ANGPTL4 in the fat cells of mice. The experiments show that when mice are exposed to cold, the levels of ANGPTL4 decrease in the brown fat cells. This allows the activity of lipoprotein lipase to increase so that these cells are able to take up more fat molecules.

However, the opposite happens in white fat cells during cold exposure. The levels of ANGPTL4 increase, which decreases the activity of lipoprotein lipase in white fat cells to allow fat molecules to be shuttled specifically to the brown fat cells. Further experiments suggest that the opposite regulation of ANGPTL4 in brown and white fat cells could be due to a protein called AMPK. This protein is found at higher levels in brown fat cells than in white fat cells and is produced by brown fat cells during cold exposure.

Taken together, Dijk et al. show that organs and cells work together to ensure that fat molecules are appropriately distributed to cells in need of energy, such as to brown fat cells during cold. How these findings could be used to stimulate fat consumption by brown fat cells in humans remains open for further investigation.

*Nedergaard, 2004*). Uncoupling in BAT is mediated by the uncoupling protein UCP1, which is highly abundant specifically in BAT (*Cannon and Nedergaard, 2004*). Studies in the last decade have shown the presence of BAT in humans and have provided preliminary evidence for an inverse relationship between BAT activity and parameters of obesity (*van Marken Lichtenbelt et al., 2009*; *Virtanen et al., 2009*; *Wang et al., 2011*). As a consequence, interest in BAT function and the possible targeting of BAT for treatment or prevention of metabolic diseases has surged.

Upon cold exposure, oxidation of fuels by BAT is dramatically increased. In addition to circulating glucose and free fatty acids, fatty acids derived from circulating triglyceride-rich lipoproteins (TRLs) represent a major fuel source for BAT (*Cannon and Nedergaard, 2004*). The liberation of fatty acids from TRLs is catalyzed by the enzyme lipoprotein lipase (LPL), which is highly abundant in BAT (*Bartelt et al., 2011*; *Kersten, 2014*). Cold exposure markedly stimulates LPL activity in BAT, causing a concomitant increase in TRL-derived fatty acid uptake and even uptake of whole lipoprotein particles (*Bartelt et al., 2011*; *Khedoe et al., 2014*; *Klingenspor et al., 1996*). The increase in fatty acid uptake upon cold exposure, which can be mimicked by pharmacological ß3-adrenergic receptor activation, is highly specific for BAT, suggesting that the body may specifically re-direct lipid fuels to BAT during cold exposure (*Bartelt et al., 2011*; *Berbée et al., 2015*; *Khedoe et al., 2014*). Both transcriptional and (post-)translational regulation has been implicated in the increased LPL activity in BAT upon cold exposure. However, the specific underlying mechanisms have remained elusive (*Carneheim et al., 1988*; *Giralt et al., 1990*; *Mitchell et al., 1992*).

Angiopoietin-like 4 (ANGPTL4) has previously been identified as an inhibitor of LPL activity in muscle and WAT. Alterations of *Angptl4* expression in these tissues mediate the changes in LPL activity observed during exercise and fasting, respectively (*Catoire et al., 2014*; *Kroupa et al., 2012*). In the initial paper describing the cloning of *Angptl4*, we had seen high expression of *Angptl4* mRNA in BAT (*Kersten et al., 2000*). Since the exact mechanisms behind regulation of LPL activity in BAT upon cold exposure are currently unclear, we hypothesized that ANGPTL4 may act as an important regulator of LPL-mediated fatty acid uptake into BAT. Accordingly, in the present paper we studied the role of ANGPTL4 in lipid metabolism during cold exposure, taking advantage of *Angptl4*-deficient (*Angptl4*-/-) mice and *Angptl4*-overexpressing (*Angptl4*-Tg) mice.

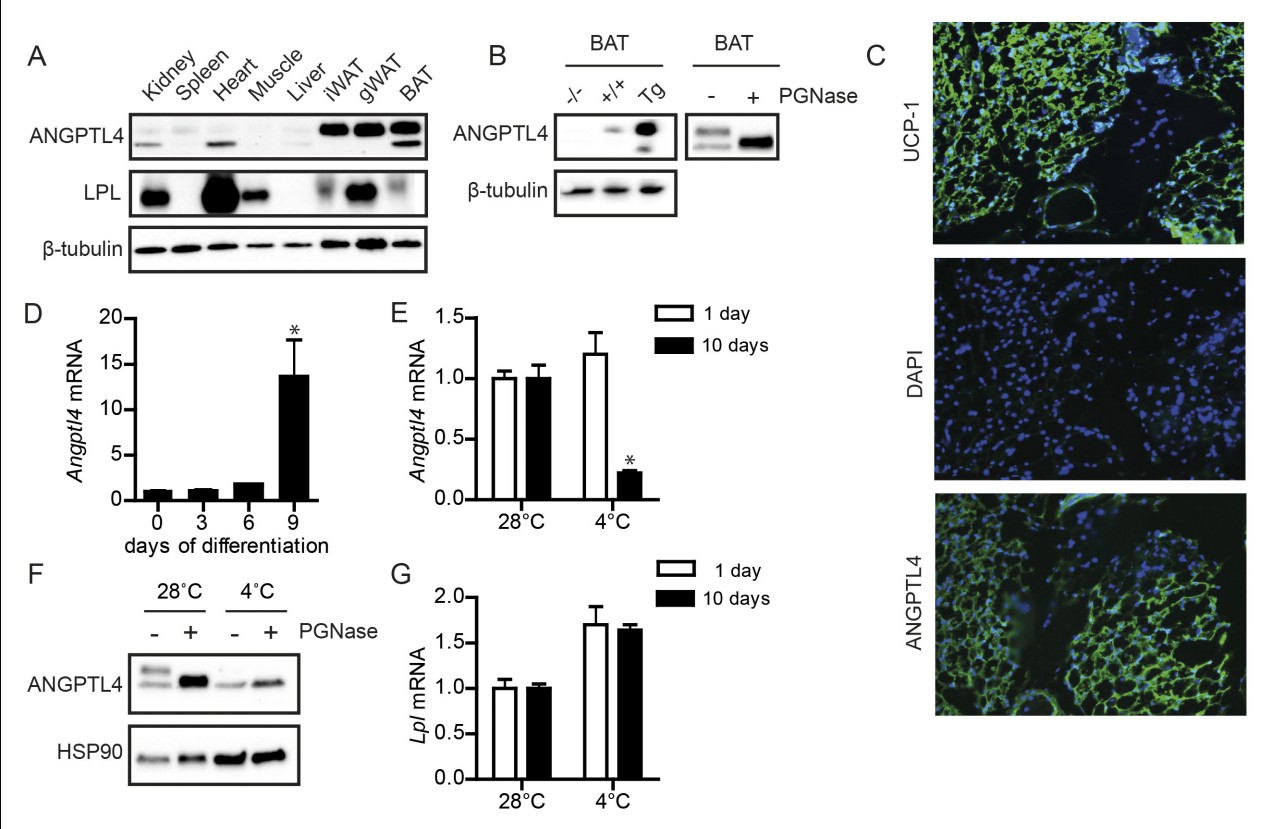

**Figure 1.** ANGPTL4 expression in BAT is down-regulated upon sustained cold exposure. (A) Immunoblot for mouse ANGPTL4 and mouse LPL in lysates of kidney, spleen, heart, muscle, gonodal WAT, inguinal WAT and BAT of a C57BL/6J wild-type mouse. (B) Validation of anti-mANGPTL4 antibody in BAT lysates of *Angptl4-/-*, wild-type and *Angptl4*-Tg mice (*left panel*). Detection of glycosylated and non-glycosylated mANGPTL4 following treatment of BAT homogenate of a wild-type mouse with PGNase (*right panel*). (C) Immunofluorescent staining of UCP1 (*upper panel*; UCP1 = green, DAPI = blue), DAPI only (*middle panel*) and hANGPTL4 (*lower panel*; hANGPTL4 = green, DAPI = blue) in frozen sections (5 μm) of human BAT. (D) *Angptl4* mRNA in T37i cells after 0, 3, 6 or 9 days of differentiation. (E) *Angptl4* mRNA in BAT lysates of wild-type mice exposed to 4°C or 28°C for 1 or 10 days. (F) Immunoblot for ANGPTL4 in BAT homogenates of wild-type mice exposed to 4°C or 28°C for 10 days, following treatment with or without PGNase. (G) *Lpl* mRNA in BAT lysates of wild-type mice exposed to 4°C or 28°C for 1 or 10 days. * Statistically significant compared to control wells or compared to mice exposed to 28°C according to Student's t-test (p<0.05). Error bars represent ± SEM. n = 8–10 mice per group.

# Results

## ANGPTL4 expression in BAT is down-regulated upon sustained cold exposure

ANGPTL4 and LPL protein are co-expressed in BAT (*Figure 1A*), suggesting a possible role for ANGPTL4 in regulating LPL in BAT. The specificity of the antibody used in the immunoblotting for ANGPTL4 was demonstrated by the lack of observable signal in BAT of *Angptl4-/-* mice (*Figure 1B* - left panel). Interestingly, we detected bands for ANGPTL4 at a molecular weight of ~52 kDa and ~45 kDa. The expected molecular weight of ANGPTL4 protein is 45 kDa, with a predicted glycosylation site at amino acid Asparagine-177 (in humans) or Asparagine 181 (in mice) (*Ge et al., 2004*; *Kim et al., 2000*; *Yang et al., 2008*; *Yoon et al., 2000*). Consistent with the notion that these bands represent glycosylated (~52 kDa) and non-glycosylated forms (~45 kDa) of ANGPTL4, the 52 kDa band disappeared following treatment of BAT lysates with the endoglycosidase PGNase, while the 45 kDa band became more intense (*Figure 1B* – right panel). These data indicate that ANGPTL4 protein is present in BAT in glycosylated and non-glycosylated forms. Immunofluorescent staining on sections from human BAT obtained during surgery (as qualified based on UCP1 staining) showed that ANGPTL4 is also expressed in human BAT (*Figure 1C*). In agreement with high ANGPTL4

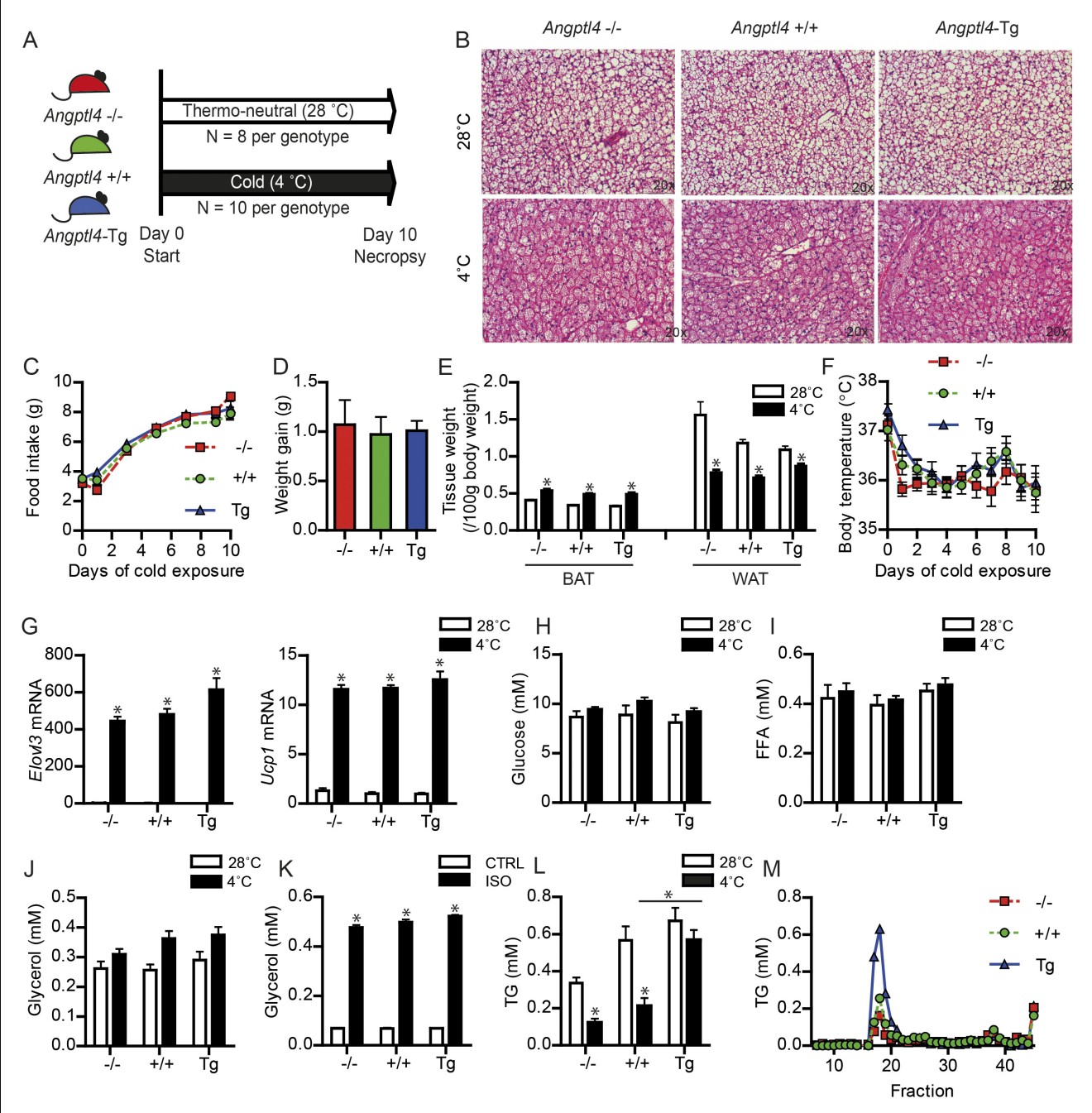

**Figure 2.** Down-regulation of ANGPTL4 in BAT upon sustained cold exposure affects plasma TG levels. (**A**) Schematic representation of cold exposure experiment with *Angptl4-/-*, wild-type and *Angptl4-*Tg mice. (**B**) Haematoxylin & Eosin staining on BAT sections (5 μm) of *Angptl4-/-*, wild-type and *Angptl4-*Tg mice exposed to 4°C or 28°C for 10 days. (**C**) Food intake of *Angptl4-/-*, wild-type and *Angptl4-*Tg mice exposed to 4°C for 10 days. (**D**) Weight gain of *Angptl4-/-*, wild-type and *Angptl4-*Tg mice exposed to 4°C during 10 days. (**E**) BAT and WAT tissue weights and (**F**) body temperature of *Angptl4-/-*, wild-type and *Angptl4-*Tg mice exposed to 4°C or 28°C for 10 days. (**G**) *Elovl3* and *Ucp1* mRNA expression levels of *Angptl4-/-*, wild-type and *Angptl4-*Tg mice exposed to 4°C or 28°C for 10 days. (**H**) Plasma glucose, (**I**) plasma free fatty acids, and (**J**) plasma glycerol levels of *Angptl4-/-*, wild-type and *Angptl4-*Tg mice exposed to 4°C or 28°C for 10 days. (**K**) Glycerol levels in medium of differentiated primary white adipocytes from *Angptl4-/-*, wild-type and *Angptl4-*Tg mice, serum-starved and treated with 10 μM isoproterenol for 3 hr. (**J**) plasma TG levels of *Angptl4-/-*, wild-type and *Angptl4-*Tg mice exposed to 4°C or 28°C for 10 days. (**K**) Fast protein liquid chromatography (FPLC) on pooled plasma samples of *Angptl4-/-*, wild-type and *Angptl4-*Tg mice exposed to 4°C for 10 days, followed by analysis of TG levels in all fractions. *Statistically significant compared to mice of equal genotype at 28°C or between groups as indicated by bars according to two-way ANOVA followed by a post-hoc Tukey HSD test (p<0.05). Error bars represent ± SEM. *n* = 8–10 mice per group.

expression levels in BAT, *Angptl4* mRNA increased markedly upon differentiation of a mouse brown adipocyte cell line (*Figure 1D*).

We next explored the possible impact of cold on ANGPTL4 expression in BAT. Intriguingly, whereas 1 day of cold exposure did not affect *Angptl4* expression, 10 days of cold exposure led to a marked reduction in *Angptl4* mRNA (*Figure 1E*). The reduction in *Angptl4* mRNA was paralleled by a marked decrease in ANGPTL4 protein, particularly the glycosylated form of ANGPTL4 (*Figure 1F*). In contrast, expression of *Lpl* was mildly elevated after both 1 and 10 days of cold exposure (*Figure 1G*).

## Down-regulation of ANGPTL4 expression promotes BAT LPL activity upon sustained cold exposure.

To investigate a possible role for ANGPTL4 in BAT function, we exposed *Angptl4*-/-, wild-type and *Angptl4*-Tg mice to a cold (4°C) or thermo-neutral (28°C) environment for 10 days in order to activate and recruit BAT (*Figure 2A*). The *Angptl4*-Tg mice overexpress *Angptl4* under its own promoter and show elevated *Angptl4* expression in a variety of tissues, including BAT (*Mandard et al., 2006*). The sustained cold exposure resulted in pronounced changes in BAT morphology, food intake, body weight, weights of BAT and WAT, and body temperature, but no clear differences between the genotypes could be observed (*Figure 2B–F*). Likewise, expression of the key thermogenic genes *Ucp1* and *Elovl3* was significantly increased upon cold exposure but not affected by *Angptl4* genotype (*Figure 2G*).

Following cold exposure, the energy requirements of BAT increase dramatically. Two major fuel sources for BAT are plasma glucose and free fatty acids, both of which were unaltered by cold and *Angptl4* genotype (*Figure 2H,I*). Also, plasma glycerol, which is perhaps a better indicator of adipose tissue lipolysis than free fatty acids, was not different between the three genotypes (*Figure 2J*). Additionally, *ex vivo* treatment of differentiated primary white adipocytes with the non-selective β-adrenergic receptor agonist isoproterenol indicated a lack of effect of *Angptl4* genotype on adipose tissue lipolysis (*Figure 2K*). Besides glucose and free fatty acids, circulating TG represent a major fuel for BAT during cold (*Cannon and Nedergaard, 2004*). Based on the marked decrease in *Angptl4* mRNA levels in BAT upon prolonged cold exposure, we hypothesized that ANGPTL4 might play a role in the metabolism of circulating TG in BAT during cold. In line with this notion, the reduction in plasma TG visible in wild-type and *Angptl4*-/- mice in response to cold was greatly attenuated in *Angptl4*-Tg mice (*Figure 2L*). Lipoprotein profiling by fast protein liquid chromatography (FPLC) supported markedly augmented plasma TRL levels in cold-exposed *Angptl4*-Tg mice (*Figure 2M*).

It is well-established that cold-induced reductions in plasma TG levels are mediated by increased LPL activity in BAT (*Bartelt et al., 2011*; *Kersten, 2014*; *Klingenspor et al., 1996*). Accordingly, we tested whether ANGPTL4 levels influence changes in BAT LPL activity upon prolonged cold exposure. Whereas wild-type mice respond to cold with a reduction in BAT ANGPTL4 mRNA and protein levels, *Angptl4*-Tg mice maintain higher ANGPTL4 mRNA and protein levels (*Figure 3A,B*). Mirroring the levels of ANGPTL4, the cold-induced changes in LPL activity in BAT exhibited a gradient across the three *Angptl4* genotypes, with highest LPL activity observed in *Angptl4*-/- mice and lowest LPL activity in *Angptl4*-Tg mice (*Figure 3C*). Furthermore, the marked increase in LPL activity during cold observed in the wild-type mice was significantly blunted in *Angptl4*-Tg mice (*Figure 3C*). Interestingly, the two-fold increase in LPL activity by cold in *Angptl4*-/- mice indicates that part of the induction of LPL activity in BAT is independent of ANGPTL4 (*Figure 3C*), possibly via an increase in *Lpl* mRNA, which was observed in all three genotypes (*Figure 3D*) (*Bartelt et al., 2011*; *Klingenspor et al., 1996*). Overall, the marked gradient in LPL activity between *Angptl4*-/-, wild-type, and *Angptl4*-Tg mice strongly suggests that ANGPTL4 acts as an inhibitor of LPL activity in BAT.

## Down-regulation of ANGPTL4 expression promotes uptake of TRL-derived fatty acids by BAT upon sustained cold exposure

To investigate the role of ANGPTL4 in plasma TG clearance by BAT, we injected cold-exposed *Angptl4*-/-, wild-type and *Angptl4*-Tg mice with radiolabelled VLDL-like emulsion particles containing glycerol tri[³H]oleate (hydrolysable by LPL; TRL-derived fatty acids) and [¹⁴C]cholesteryl-oleate

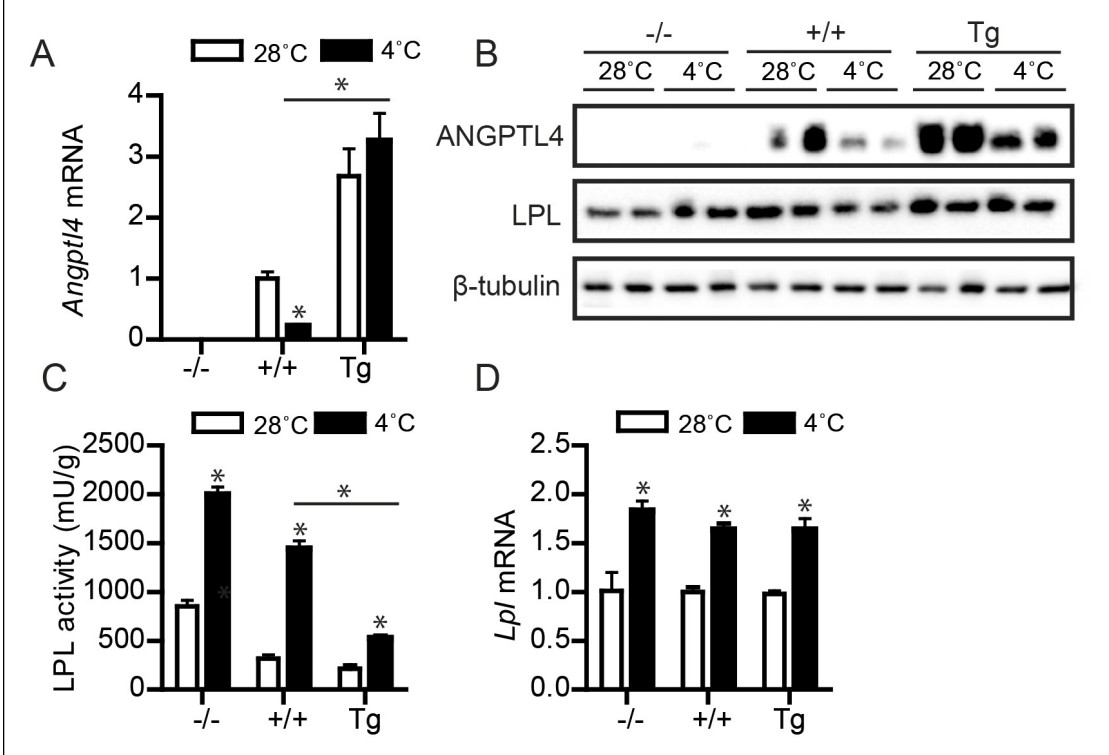

**Figure 3.** Down-regulation of ANGPTL4 in BAT upon sustained cold exposure promotes an increase in BAT LPL activity. (A) *Angptl4* mRNA in BAT of *Angptl4*-/-, wild-type and *Angptl4*-Tg mice exposed to 4°C or 28°C for 10 days. (B) Immunoblot for ANGPTL4 and LPL in BAT homogenates from *Angptl4*-/-, wild-type and *Angptl4*-Tg mice exposed to 4°C or 28°C for 10 days. (C) Total LPL activity and (D) *Lpl* mRNA in BAT of *Angptl4*-/-, wild-type and *Angptl4*-Tg mice exposed to 4°C or 28°C for 10 days. * Statistically significant compared to mice of equal genotype at 28°C or between groups as indicated by bars, according to two-way ANOVA followed by a post-hoc Tukey HSD test (p<0.05). Error bars represent ± SEM. n = 8–10 mice per group.

(not hydrolysable by LPL; TRL-Chol) (see *Figure 4—figure supplement 1* for experimental set-up) (*Rensen et al., 1995*). After 15 min, the mice were sacrificed and the tissue distribution of [3]H and [14]C activity was determined. As expected, cold exposure markedly increased the rate of clearance of the injected VLDL-like particles from the plasma (*Figure 4A,B*). However, after cold exposure, plasma clearance of glycerol tri[3H]oleate (TRL FA), but not [14C]cholesteryl-oleate (TRL Chol), was significantly slower in *Angptl4*-Tg mice, indicating that *Angptl4*-overexpression inhibits LPL-mediated plasma TG clearance in the cold (*Figure 4C,D*). Cold exposure caused a marked increase in TRL-derived fatty acid uptake into BAT in all three genotypes, indicating that part of the increase in fatty acid uptake is independent of ANGPTL4 (*Figure 4E*). However, similar to LPL activity, a clear gradient in TRL-derived fatty acid uptake into BAT was observed between the three genotypes, with lowest uptake in *Angptl4*-Tg mice (*Figure 4E*). Similar results were obtained for [14C]cholesteryl-oleate, showing a marked decrease in BAT uptake in the *Angptl4*-Tg mice (*Figure 4F*). We repeated the plasma TG clearance studies using radiolabelled lipoprotein particles with a larger diameter, resembling postprandial chylomicrons. Again, *Angptl4*-overexpression markedly reduced uptake into BAT of the TRL-derived fatty acids and the core label cholesteryl-oleyloleate (*Figure 4G,H*). To visualize the TRL uptake process, we injected hydrophobic fluorescent nanocrystals embedded in lipoprotein particles (QD-TRLs) into cold-exposed *Angptl4*-/-, wild-type and *Angptl4*-Tg mice (*Bruns et al., 2009*). In agreement with a role for ANGPTL4 in TRL processing in BAT, increased accumulation of QD-TRLs was observed after cold exposure of wild-type mice, but not of *Angptl4*-Tg mice (*Figure 4I*). Furthermore, *Angptl4*-/- mice show an accumulation of QD-TRLs in BAT even when maintained at 28°C (*Figure 4I*). Together, these data are supportive of a major role for ANGPTL4 as a regulator of LPL activity and concomitant uptake of fatty acids into BAT upon prolonged cold exposure.

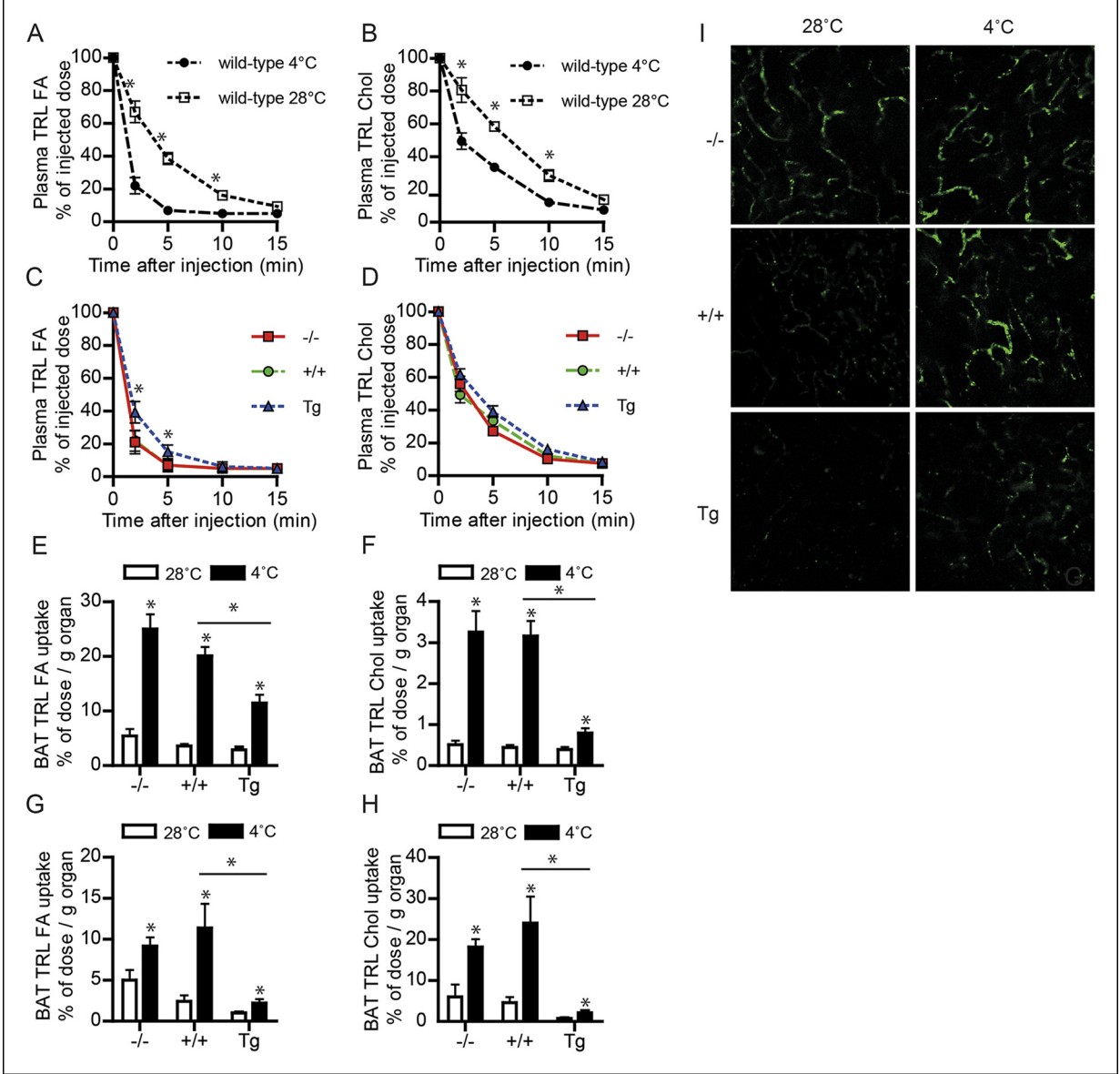

**Figure 4.** Down-regulation of ANGPTL4 in BAT upon sustained cold exposure promotes an increase in TRL-derived fatty acid uptake by BAT. (A,B) Plasma [3]H (A) and [14]C (B) activity in wild-type mice exposed to 4°C or 28°C for 10 days intravenously injected with VLDL-like particles labelled with glycerol tri[3H]oleate (TRL FA) and [14C]cholesteryl-oleate (TRL Chol). (C,D) Plasma [3]H (C) and [14]C (D) activity in *Angptl4*-/-, wild-type and *Angptl4*-Tg mice intravenously injected with VLDL-like emulsion particles labelled with glycerol tri[3H]oleate (TRL FA) and [14C]cholesteryl-oleate (TRL Chol), following exposure to 4°C for 10 days. (E,F) [3]H activity (E) and [14]C activity (F) in interscapular BAT of *Angptl4*-/-, wild-type and *Angptl4*-Tg mice exposed to 4°C or 28°C for 10 days and intravenously injected with VLDL-like particles labelled with glycerol tri[3H]oleate (TRL FA) and [14C]cholesteryl-oleate (TRL Chol). (G,H) [14]C and [3]H activity in interscapular BAT of *Angptl4*-/-, wild-type and *Angptl4*-Tg mice exposed to 4°C or 28°C for 10 days and intravenously injected with chylomicron-like particles labelled with glycerol tri[14C]oleate (TRL FA) (G) and [3H]cholesteryl-oleyloleate (TRL Chol). (I) Fluorescent image of uptake of intravenously injected QD-TRLs into BAT of *Angptl4*-/-, wild-type and *Angptl4*-Tg mice exposed to 4°C or 28°C for 9 days. Image was taken 12 min post-injection. *n* = 2 mice per group.* Statistically significant compared to mice of equal genotype at 28°C or between groups as indicated by bars, according to two-way ANOVA followed by a post-hoc Tukey HSD test (p<0.05). Error bars represent ± SEM. *n* = 7 mice per group, unless otherwise indicated.

The following figure supplement is available for figure 4:

**Figure supplement 1.** (A,B) Schematic representation of clearance studies in *Angptl4*-/-, wild-type and *Angptl4*-Tg mice, consisting of 10 days of cold exposure or thermo-neutrality, followed by measurement of triglyceride clearance via injection of either radiolabelled VLDL-like particles or chylomicron-like particles.

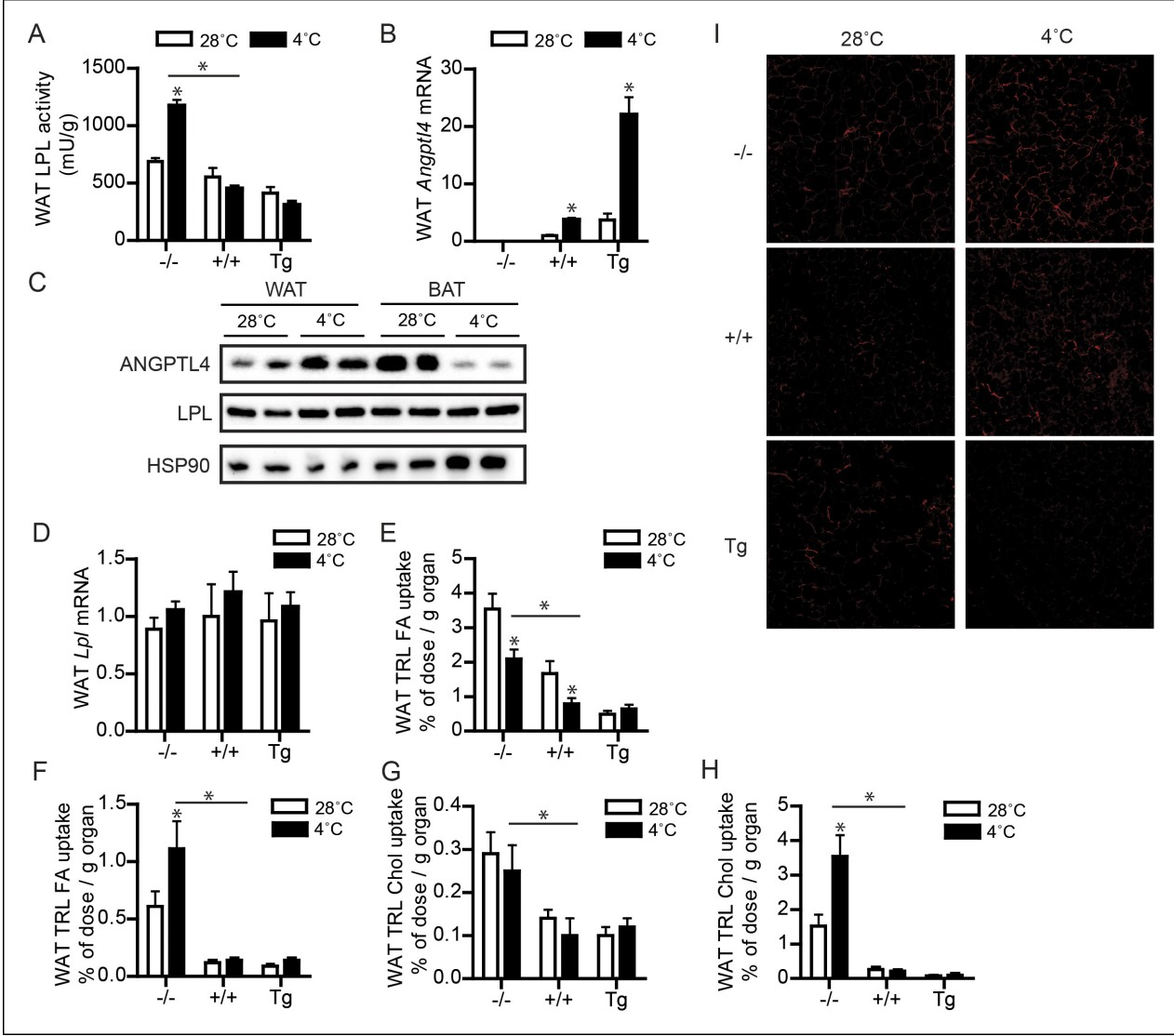

**Figure 5.** Up-regulation of ANGPTL4 in WAT upon sustained cold exposure suppresses WAT LPL activity and TRL-derived fatty acid uptake. (A) Total LPL activity levels and (B) *Angptl4* mRNA in WAT of *Angptl4*-/-, wild-type and *Angptl4*-Tg mice exposed to 4°C or 28°C for 10 days. (C) Immunoblot for ANGPTL4 and LPL in WAT homogenates of wild-type mice exposed to 4°C or 28°C for 10 days. (D) *Lpl* mRNA in WAT of *Angptl4*-/-, wild-type and *Angptl4*-Tg mice exposed to 4°C or 28°C for 10 days. (E–H) Activity of $^3$H and $^{14}$C radiolabels in WAT of *Angptl4*-/-, wild-type and *Angptl4*-Tg mice exposed to 4°C or 28°C for 10 days and intravenously injected with radiolabelled VLDL-like (E,G) and chylomicron-like particles (F,H). TRL FA uptake (E, F) reflects uptake of glycerol tri[$^3$H/$^{14}$C]oleate, whereas TRL Chol uptake (G,H) reflects uptake of the core labels [$^{14}$C]cholesteryl-oleate or [$^3$H] cholesteryl-oleyloleate. (I) Fluorescent image of uptake of intravenously injected QD-TRLs into WAT of *Angptl4*-/-, wild-type and *Angptl4*-Tg mice exposed to 4°C or 28°C for 9 days. Image was taken after perfusion of mice with PBS containing 50 IU/mL heparin and upon cryosectioning of tissues. *n* = 2 mice per group. * Statistically significant compared to mice of equal genotype at 28°C or between groups as indicated by bars, according to two-way ANOVA followed by a post-hoc Tukey HSD test (p<0.05). Error bars represent ± SEM. *n* = 7–10 mice per group, unless otherwise indicated.

The following figure supplements are available for figure 5:

**Figure supplement 1.** (A,B) Plasma ANGPTL4 levels in 10 obese (A) and 10 lean (B) individuals before and after exposure to a mild cold (16°C) for 48 hr (*Wijers et al., 2010*).

**Figure supplement 2.** (A) Total LPL activity levels, (B) *Angptl4* mRNA, and (C) *Lpl* mRNA in inguinal WAT (iWAT) of *Angptl4*-/-, wild-type and *Angptl4*-Tg mice exposed to 4°C or 28°C for 10 days.

## ANGPTL4 expression and LPL activity are oppositely regulated in WAT and BAT upon sustained cold exposure.

Whereas BAT utilizes lipids to fuel thermogenesis, WAT provides lipid fuels to be used by BAT. Accordingly, lipid uptake and LPL activity are expected to be regulated differently in BAT as compared to WAT. Indeed, whereas in wild-type mice LPL activity in BAT increased dramatically during cold, LPL activity in WAT remained unchanged, at least in wild-type and *Angptl4*-Tg mice (*Figure 5A*). However, LPL activity was increased by cold in *Angptl4*-/- mice, indicating that ANGPTL4 prevents LPL activity in WAT from going up during cold (*Figure 5A*). In support of this function for ANGPTL4, cold exposure caused a marked increase in *Angptl4* mRNA and protein levels in WAT (*Figure 5B,C*). In contrast to *Angptl4, Lpl* mRNA levels were not significantly altered in WAT upon sustained cold exposure (*Figure 5D*). Interestingly, exposure of human subjects to mild cold (16°C) for 48 hr significantly increased plasma ANGPTL4 levels in obese subjects, but not in lean subjects. Considering the higher adipose tissue mass in obese individuals, these data suggest that the increase in ANGPTL4 production in WAT dominates ANGPTL4 levels in human plasma (*Figure 5—figure supplement 1*).

To determine if the suppressive effect of ANGPTL4 on LPL activity had any impact on uptake of TRL-derived fatty acids in WAT during cold, we measured fatty acid uptake following injection of radiolabelled VLDL-like and chylomicron-like emulsion particles. Even though the uptake behaviour of the two types of TRL-particles was somewhat different, ANGPTL4 expression caused a clear dose-dependent reduction in uptake of TRL-derived fatty acids and the core labels cholesteryl-oleate/cholesteryl-oleyloleate (*Figure 5E–H*). In addition, inhibition of TRL processing by ANGPTL4 in WAT was visually confirmed by injection of QD-TRLs, showing increased accumulation of QD-TRLs in WAT of *Angptl4*-/- mice, as compared to wild-type and *Angptl4*-Tg mice (*Figure 5I*).

It is noteworthy that in the inguinal fat depot, cold-induced changes in *Lpl* mRNA, *Angptl4* mRNA and LPL activity were similar to the changes observed in the gonadal fat depot described above, whereas changes in uptake of TRL-derived fatty acids upon cold exposure were quite distinct, most likely due to marked activation of browning in inguinal fat (*Figure 6A,B*; *Figure 5—figure supplement 2*). In contrast to BAT and WAT, uptake of TRL-derived fatty acids or cholesteryl-oleate/cholesteryl-oleyloleate from VLDL-like particles and chylomicron-like particles was minimally different between *Angptl4*-/-, wild-type and *Angptl4*-Tg mice in liver, skeletal muscle and spleen (*Figure 6A, B*). Taken together, our data indicate that in BAT down-regulation of ANGPTL4 promotes uptake of plasma TRL-derived fatty acids via enhanced LPL activity, whereas in WAT up-regulation of ANGPTL4 suppresses uptake of plasma TRL-derived fatty acids via inhibition of LPL activity, thereby directing plasma TG to BAT to be used as fuel.

## The opposite regulation of ANGPTL4 expression in BAT and WAT may be mediated by differential activation of AMPK

We next explored the mechanism accounting for the inverse regulation of ANGPTL4 expression in BAT and WAT. Since many of the cold-induced adaptations are triggered by β-adrenergic signalling, we examined the effect of β-adrenergic activation on ANGPTL4 expression in BAT and WAT using murine BAT (T37i cells) and WAT (3T3-F442a) cell lines and primary cells (*Zennaro et al., 1998*). In both BAT and WAT cells, treatment with the non-selective β-receptor agonist isoproterenol consistently resulted in a marked increase in expression of *Angptl4* mRNA (*Figure 7A,B*). Induction of ANGPTL4 by β-adrenergic stimulation was confirmed at the protein level (*Figure 7C*). From these data, it is evident that activation of β-adrenergic signalling may contribute to the cold-induced up-regulation of ANGPTL4 in WAT, but cannot explain the down-regulation of ANGPTL4 observed in BAT. Levels of *Lpl* mRNA and protein in white and brown adipocytes were only mildly affected by treatment with β-adrenergic agonists (*Figure 7C,D*).

We therefore considered mechanisms that are clearly different between BAT and WAT. Previously, it had been shown that 5'AMP-activated protein kinase (AMPK) is progressively activated with prolonged cold exposure and that AMPK expression is significantly higher in BAT compared to WAT (*Mulligan et al., 2007*). Consistent with this finding, we found that the catalytic α-subunits of AMPK, as well as phosphorylation of AMPK at Threonine-172 – indicative of AMPK activity – are barely detectable in WAT in the basal state (*Figure 7E*) (*Viollet et al., 2009*, *2010*). Previously, differential tissue expression of isoforms of the α, β, and γ subunits of the AMPK heterotrimer was

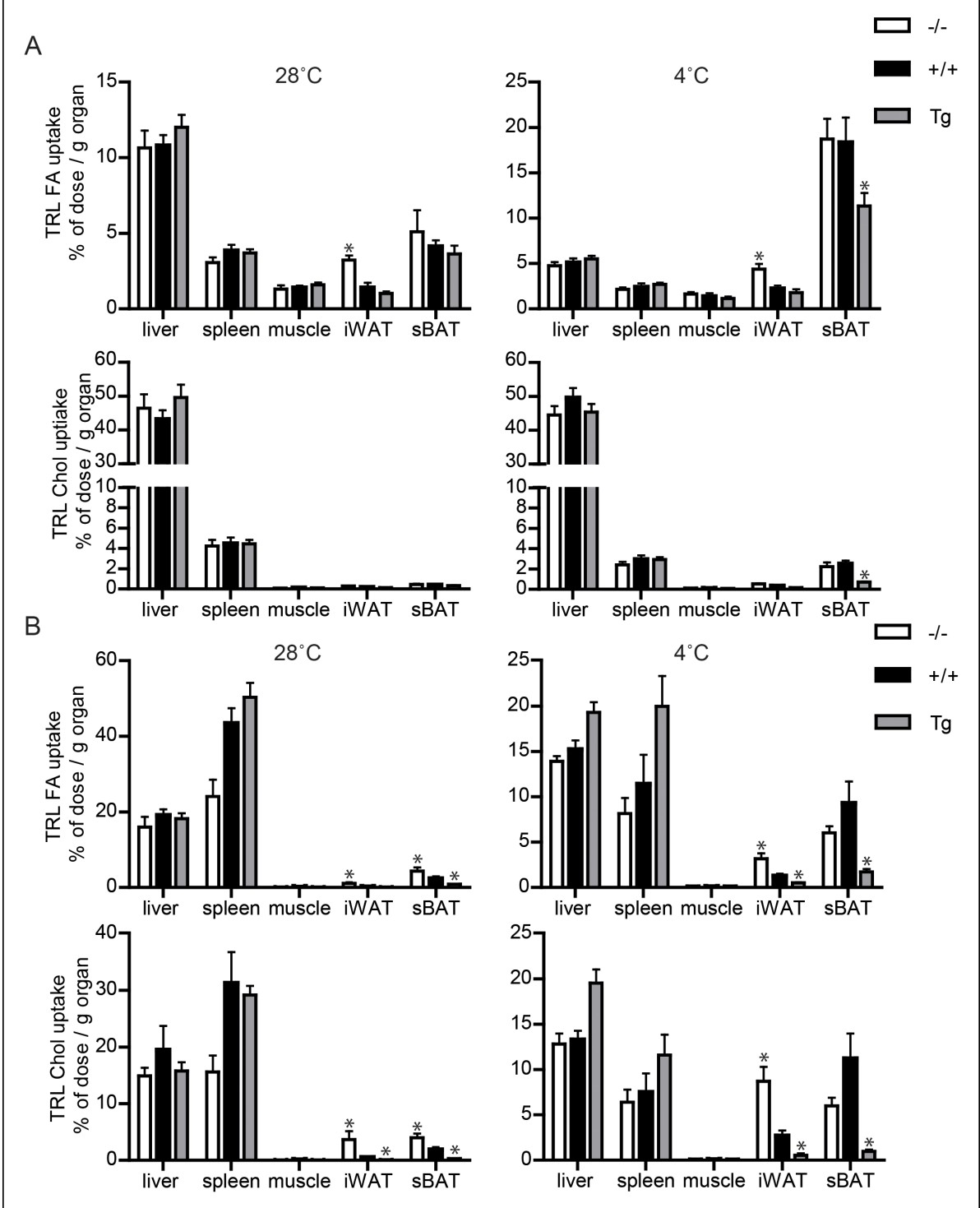

**Figure 6.** Uptake of TRL-like particles in liver, spleen and muscle is not affected by *Angptl4* genotype. (**A**) [3]H and [14]C activity in liver, spleen, muscle, inguinal WAT (iWAT) and subscapular BAT (sBAT) of *Angptl4-/-*, wild-type and *Angptl4-*Tg mice exposed to 4°C or 28°C for 10 days and intravenously injected with VLDL-like emulsion particles labelled with glycerol tri[3H]oleate (TRL FA) and [14C]cholesteryloleate (TRL Chol). (**B**) [3]H and [14]C activity in liver, spleen, muscle, iWAT and sBAT of *Angptl4-/-*, wild-type and *Angptl4-*Tg mice exposed to 4°C or 28°C for 10 days and intravenously injected with chylomicron-like particles labelled with glycerol tri[14C]oleate (TRL FA) and [3H]cholesteryl-oleyloleate (TRL Chol). *Statistically significant compared to values of wild-type mice according to Student's t-test (p<0.05). Error bars represent ± SEM. *n* = 7 mice per group.

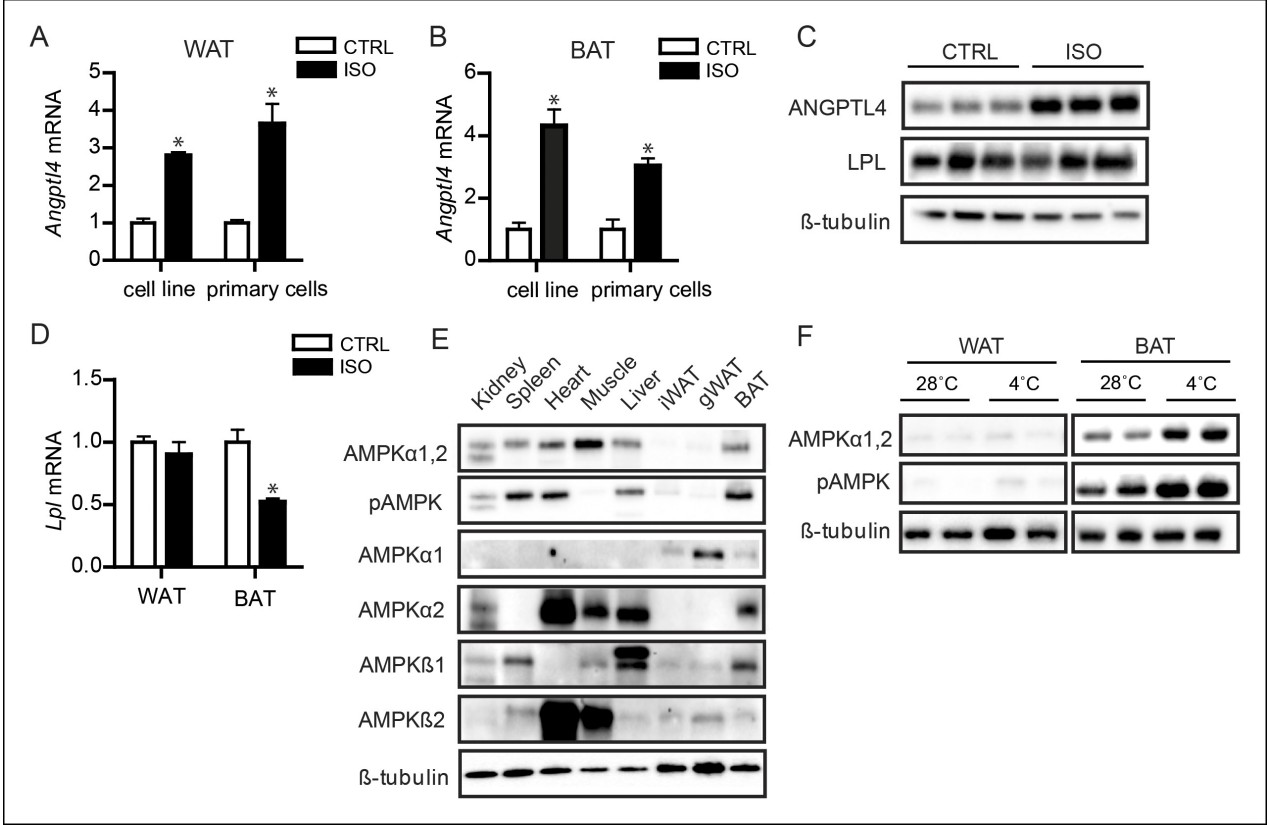

**Figure 7.** AMPK is activated in BAT, but not WAT, upon sustained cold exposure. (**A**) *Angptl4* mRNA in differentiated mouse white adipocytes (primary adipocytes and 3T3F442a adipocytes) upon treatment with 10 µM isoproterenol (ISO) or control (CTRL) for 3 hr. (**B**) *Angptl4* mRNA in differentiated brown adipocytes (primary adipocytes and T37i adipocytes) treated with 10 µM isoproterenol (ISO) orcontrol (CTRL) for 3 hr. (**C**) Immunoblot for ANGPTL4 and LPL protein in differentiated 3T3F422a cells treated with 10 µM isoproterenol (ISO) or control (CTRL) for 3 hr. (**D**) *Lpl* mRNA in differentiated mouse primary white or brown adipocytes upon treatment with 10 µM isoproterenol (ISO) or control (CTRL) for 3 hr. (**E**) Immunoblot for AMPKα1,2 and phospho-AMPK Thr172, AMPKα1, AMPKα2, AMPKβ1 and AMPKβ2 in tissue lysates of kidney, spleen, heart, muscle, liver, inguinal WAT, gonodal WAT and BAT. Homogenates are identical to the homogenates presented in *Figure 1A*. (**F**) Immunoblot for AMPKα1,2 and phospho-AMPK Thr172 in BAT and WAT lysates of wild-type mice exposed to 4°C or 28°C for 10 days. *Statistically significant compared to control samples or between indicated treatments according to Student's t-test (p<0.05). Error bars represent ± SEM.

suggested to determine cellular and systemic responses to different metabolic stressors (*Viollet et al., 2009*, *2010*). Intriguingly, we detected large differences in basal AMPK α- and β-subunit isoform distribution between BAT and WAT and, more specifically, found a high expression of the AMPKα2 catalytic subunit and the AMPKβ1 regulatory subunit in BAT compared to WAT (*Figure 7E*). The differences in total AMPK expression and subunit distribution may explain why WAT AMPK levels remain weak following sustained cold exposure, whereas a strong increase in AMPK and phospho-AMPK is observed in BAT (*Figure 7F*).

In muscle cells, we found that AMPK activation strongly down-regulates ANGPTL4 mRNA and protein levels (*Catoire et al., 2014*). To examine the consequences of AMPK activation in BAT, we treated differentiated T37i brown adipocytes with multiple AMPK activators, including AICAR, A769662, metformin, and phenformin hydrochloride. Without exception, AMPK activation markedly down-regulated *Angptl4* mRNA expression (*Figure 8A*). To further confirm the down-regulation of ANGPTL4 by AMPK, we treated different *in vitro* model systems for BAT with the AMPK activator AICAR. AICAR treatment of differentiated T37i adipocytes, BA adipocytes (*Uldry et al., 2006*) or murine primary brown adipocytes resulted in a marked decrease in ANGPTL4 mRNA and protein levels (*Figure 8B,C*). Part of the inhibitory effect of AMPK activation on *Angptl4* expression in differentiated T37i adipocytes could be rescued by siRNA-mediated knock-down of AMPKα1 and AMPKα2, corroborating the suppressive effect of AMPK on *Angptl4* expression (*Figure 8D–F*).

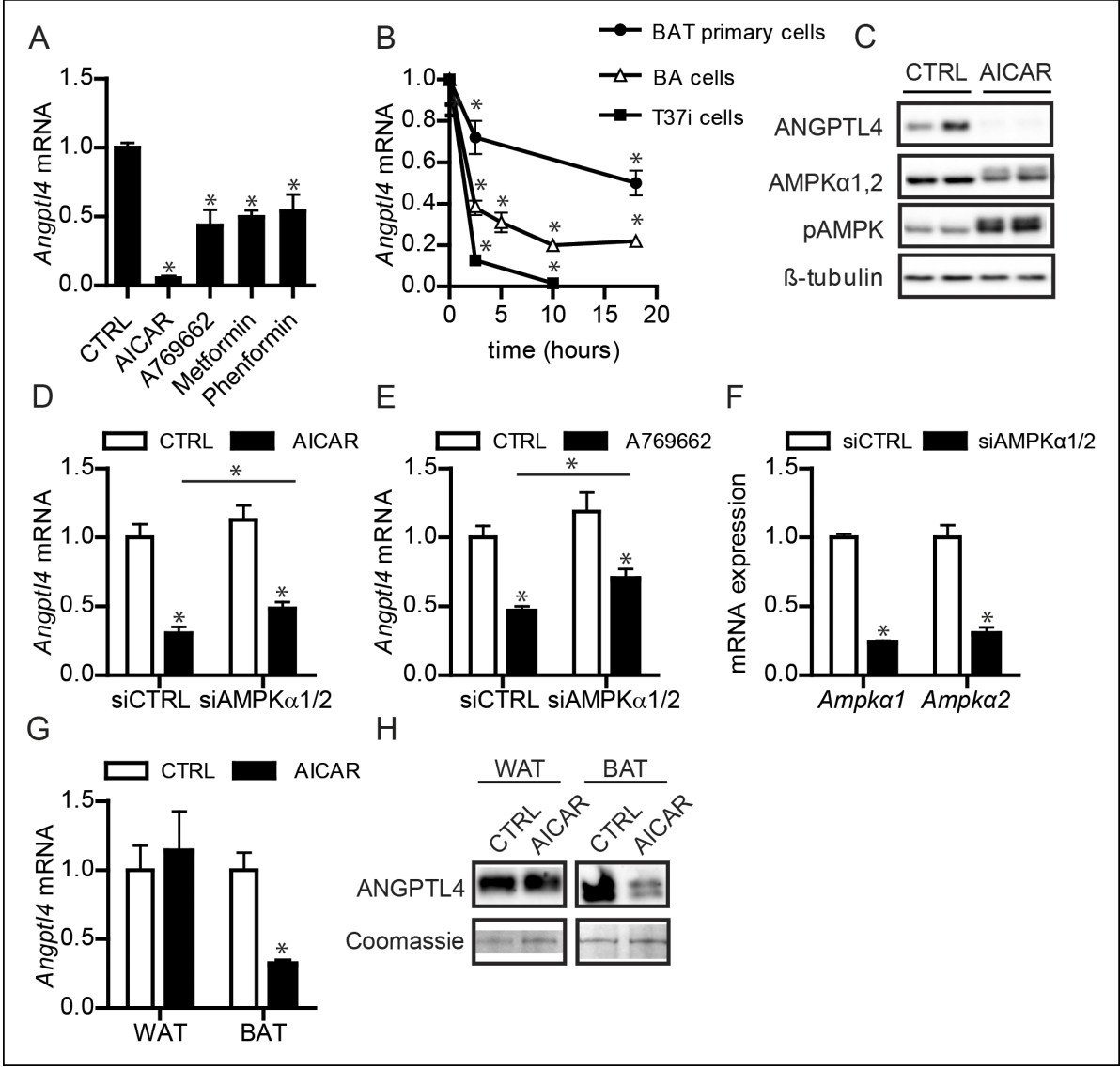

**Figure 8.** Activation of AMPK down-regulates ANGPTL4 expression specifically in brown adipocytes. (A) *Angptl4* mRNA in differentiated T37i adipocytes treated for 6 hr with 1 mM AICAR, 100 μM A769662, 1 mM metformin or 250 μM phenformin hydrochloride. (B) *Angptl4* mRNA in differentiated primary brown adipocytes, BA adipocytes, or T37i adipocytes treated for indicated times with 1 mM AICAR. (C) Immunoblot for ANGPTL4, LPL, AMPKα1,2 and phospho-AMPK Thr172 in differentiated T37i cells treated with control (CTRL) or 1 mM AICAR for 3 hr. (D) *Angptl4* mRNA in differentiated T37i adipocytes treated with CTRL siRNA or siRNA against AMPKα1 and AMPKα2 for 48 hr, followed by incubation with control (CTRL) or 1 mM AICAR for 3 hr. (E) *Angptl4* mRNA in differentiated T37i adipocytes treated with CTRL siRNA or siRNA against AMPKα1 and AMPKα2 for 48 hr, followed by incubation with H$_2$O control medium (CTRL) or 100 μM A769662 for 6 hr. (E) *Ampkα1* and *Ampkα2* mRNA in differentiated T37i adipocytes treated with CTRL siRNA or siRNA against AMPKα1 and AMPKα2 for 48 hr. (F) *Angptl4* mRNA levels in BAT and WAT explants from C57BL/6J wild-type mice (~50 μg) treated with H$_2$O control medium (CTRL) or 1 mM AICAR for 3 hr. (G) Immunoblot for ANGPTL4 in BAT and WAT explants from C57BL/6J wild-type mice (~50 mg) treated with H$_2$O control medium (CTRL) or 1 mM AICAR for 3 hr. *Statistically significant compared to control samples or between indicated treatments, according to Student's t-test (p<0.05). Error bars represent ± SEM.

Consistent with the notion that the negative regulation of ANGPTL4 by AMPK is specific for BAT, AICAR treatment markedly reduced ANGPTL4 mRNA and protein levels in BAT explants, but not WAT explants (*Figure 8G,H*). Based on these data, we propose that the different amount and activation of AMPK between BAT and WAT may be the critical factor in the differential regulation of ANGPTL4 between the two tissues during sustained cold.

## AMPK may act through PPAR to regulate ANGPTL4 levels

We further pursued the mechanism behind the down-regulation of ANGPTL4 by AMPK. Treatment of brown adipocytes with AICAR and the transcriptional inhibitor actinomycin D showed that both compounds reduced *Angptl4* gene expression by nearly the same extent. Co-treatment of brown adipocytes with actinomycin D and AICAR did not result in an additive effect, suggesting that AMPK activation almost completely inhibited *Angptl4* gene transcription (*Figure 9A*).

Expression of ANGPTL4 is under sensitive control of peroxisome proliferator-activated receptors (PPARs) in many tissues, with PPARγ being the dominant regulator of ANGPTL4 in adipose tissue (*Kersten et al., 2000*; *Mandard et al., 2004*; *Yoon et al., 2000*). We found that PPARγ agonists and to a lesser extent agonists for PPARα and PPARδ highly induce ANGPTL4 mRNA and protein in BA and T37i brown adipocytes (*Figure 9B,C*).

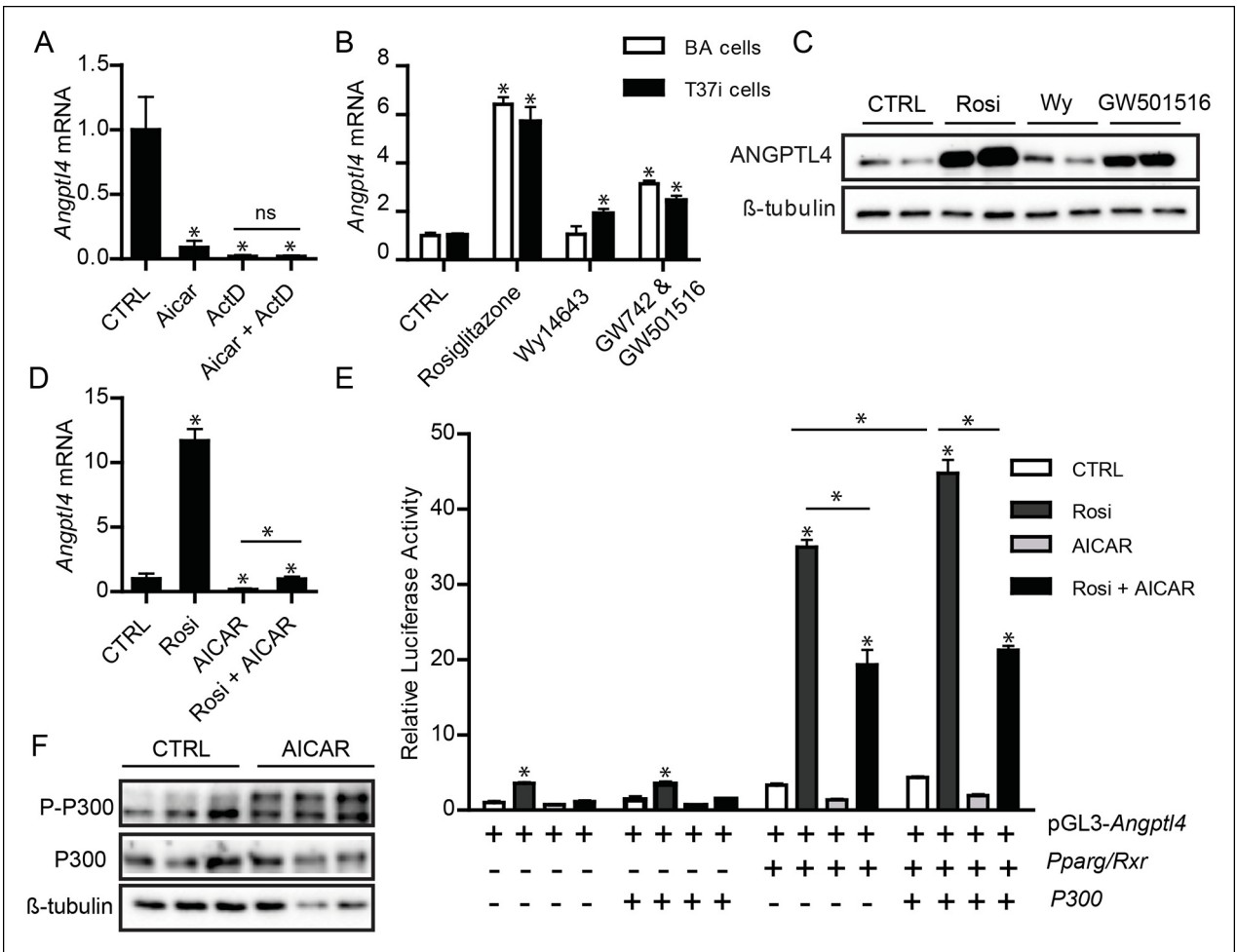

**Figure 9.** Down-regulation of *Angptl4* expression by AMPK is likely mediated via inhibition of PPARγ-mediated transcription of *Angptl4*. (**A**) *Angptl4* mRNA in differentiated T37i adipocytes pre-incubated with 0.5 μg/mL actinomycin D (ActD) or DMSO control for 1 hr and treated with 1 mM AICAR or control for 3 hr. (**B**) *Angptl4* mRNA in differentiated T37i or BA adipocytes treated with DMSO control, 5 μM rosiglitazone, 10 μM Wy14643, or 5 μM GW742 (BA adipocytes) or 5 μM GW501516 (T37i adipocytes) for 6 hr. (**C**) Immunoblot for ANGPTL4 in differentiated T37i adipocytes treated with 5 μM rosiglitazone (Rosi), 10 μM Wy14643 (Wy), and 5 μM GW501516 for 24 hr. (**D**) *Angptl4* mRNA in differentiated T37i cells treated with DMSO control, 5 μM rosiglitazone (Rosi), 1 mM AICAR or both rosiglitazone and AICAR for 6 hr. (**E**) Relative luciferase activity of HepG2 cells transfected with pGL3-*Angptl4*, pSG5-*Pparγ*, pSG5-*Rxr*, pcDNA-*P300HA* vectors, as indicated and treated 16 hr post-transfection with 5 μM rosiglitazone, 1 mM AICAR or both compounds for 9 hr. Data are represented as ± SD. (**F**) Immunoblot of P300 and phospho-P300 (Ser-89) in differentiated T37i brown adipocytes treated with control or 1 mM AICAR for 3 hr. *Statistically significant compared to control samples or between indicated samples according to Student's t-test (p<0.05). Error bars represent ± SEM, unless otherwise specified.

Previous studies have shown that AMPK may inhibit PPARα and PPARγ transcriptional activity (*Leff, 2003*; *Sozio et al., 2011*). Indeed, we observed that AMPK activation almost completely blocked the induction of *Angptl4* mRNA following treatment with the PPARγ agonist rosiglitazone (*Figure 9D*). Accordingly, we hypothesized that activation of AMPK following prolonged cold exposure may inhibit PPARγ-mediated transcription of the *Angptl4* gene. To examine this possibility, a luciferase construct was prepared containing the three conserved PPAR response elements (PPREs) of intron 3 of the murine *Angptl4* gene. These three PPREs have previously been shown to be responsible for PPAR-mediated up-regulation of *Angptl4* (*Kaddatz et al., 2010*; *Mandard et al., 2004*). In HepG2 cells transfected with the *Angptl4* PPRE construct , rosiglitazone treatment significantly induced luciferase activity, which was further increased upon co-transfection with *Pparγ*/Retinoid X receptor (*Rxr*). By contrast, co-treatment with AICAR blunted the increase in luciferase activity (*Figure 9E*). Interestingly, whereas co-transfection of the PPARγ co-activator *P300* (*Gelman et al., 1999*) further stimulated luciferase activity in the absence of AICAR, it failed to do so in the presence of AICAR, suggesting that regulation of P300 may be part of the mechanism of inhibition of ANGPTL4 by AMPK (*Figure 9E*). In support of this possibility, treatment of brown adipocytes with AICAR resulted in phosphorylation of P300 at serine residue 89, which is known to reduce the capacity of P300 to co-activate nuclear transcription factors, whereas no change in total P300 protein levels was observed (*Figure 9F*). Collectively, these data suggest that the negative regulation of ANGPTL4 in BAT upon cold exposure may be mediated by the inhibition of PPARγ transcriptional activity by AMPK.

## Discussion

The energy requirements of BAT increase manifold during cold exposure. The increased energy demands coincide with a marked increase in LPL activity, stimulating uptake of TRL-derived fatty acids (*Bartelt et al., 2011*; *Bertin et al., 1985*; *Khedoe et al., 2014*; *Klingenspor et al., 1989*, *1996*). Increased LPL activity has been shown to be essential for the lipid-lowering effect of cold exposure, as injection of heparin or tetrahydrolipstatin compromises LPL-dependent uptake of TRLs and TRL-derived fatty acids (*Bartelt et al., 2011*). Since *Lpl* mRNA in BAT is only moderately increased upon prolonged cold exposure, it has been suggested that the pronounced increase in LPL activity in BAT occurs at the post-translational level (*Giralt et al., 1990*; *Klingenspor et al., 1996*). Our data demonstrate that a substantial part of the increase in LPL activity in BAT during prolonged cold exposure is mediated by down-regulation of ANGPTL4.

Overall, our findings reveal a major role for ANGPTL4 in the regulation of lipid partitioning during sustained cold. Specifically, the data implicate ANGPTL4 as an important mediator of preferential shuttling of TRL-derived fatty acids to BAT during cold exposure. Via direct effects on local LPL activity and subsequent fatty acid uptake, the reciprocal regulation of ANGPTL4 in BAT and WAT assures an adequate fuel delivery to BAT during cold exposure. The differential regulation of ANGPTL4 and LPL between BAT and WAT leads to corresponding changes in fatty acid uptake from TRLs, with our data showing a clear dose-dependent and causal relationship between ANGPTL4 expression and TRL-derived fatty acid uptake into both tissues. The importance of ANGPTL4 in the regulation of LPL activity during cold complements the already established role of ANGPTL4 in regulation of LPL during fasting and exercise in WAT and skeletal muscle, respectively (*Catoire et al., 2014*; *Kroupa et al., 2012*). ANGPTL4 can thus be viewed as the master regulator of tissue LPL activity and fatty acid uptake during physiological conditions such as fasting, exercise and cold exposure.

Our data suggest that the opposite regulation of ANGPTL4 expression during prolonged cold between BAT and WAT may be explained by the differential expression and activation of AMPK between the two tissues. Heterotrimeric AMPK has one catalytic (α), and two regulatory (β and γ) subunits, each having distinctive isoforms with a tissue-specific distribution (*Viollet et al., 2009*, *2010*). Tissue-specific combinations of different subunit isoforms may confer tissue-specific properties to AMPK by determining subcellular localization and substrate targeting, thereby controlling cellular and systemic responses to metabolic stressors, including sustained cold (*Viollet et al., 2009*, *2010*). Indeed, with prolonged cold exposure, AMPK becomes progressively activated in BAT and only to a minor extent in WAT (*Bauwens et al., 2011*; *Mulligan et al., 2007*).

Systemic activation of AMPK has been previously shown to increase the activity of LPL in both heart and muscle and to lower plasma TG levels (*An et al., 2005*; *Bergeron et al., 2001*; *Buhl et al.,*

*2002*; *Catoire et al., 2014*; *Geerling et al., 2014*; *Ohira et al., 2009*). Furthermore, AMPK activation was found to cause a pronounced reduction in ANGPTL4 expression in muscle cells (*Catoire et al., 2014*). Similar to muscle and heart, we found the AMPKα2 catalytic subunit to be abundantly present in BAT, but not WAT (*Viollet et al., 2009*). Together, the increased AMPK activation and different AMPK subunit expression in BAT as compared to WAT may explain why the repressive effect of cold-induced AMPK activation on ANGPTL4 expression is much more pronounced in BAT than in WAT. We suggest that regulation of ANGPTL4 in WAT during cold may be dominated by activation of β-adrenergic signalling, which may explain the increase in ANGPTL4 expression observed in WAT during sustained cold exposure.

We provide evidence that regulation of ANGPTL4 by AMPK occurs at the transcriptional level, affecting PPARγ-mediated transcription of the *Angptl4* gene. ANGPTL4 has been repeatedly shown to be a highly sensitive target of all PPAR transcription factors in a variety of tissues and cells and following a variety of physiological stimuli (*Dijk and Kersten, 2014*; *Georgiadi et al., 2010*; *Kersten et al., 2000*; *Mandard et al., 2004*). Previously, PPARγ-mediated transcription has been shown to be inhibited by activation of AMPK (*Cheang et al., 2014*; *Namgaladze et al., 2013*; *Sozio et al., 2011*). A potential link between PPARγ and AMPK may be the modulation of co-activator recruitment to PPARγ by AMPK. A well-established co-activator of PPARγ that has been shown to be regulated by AMPK is P300 (*Gelman et al., 1999*; *Leff, 2003*; *Yang et al., 2001*). P300 is a key regulator of the assembly and mobilization of the transcriptional machinery by connecting transcription factors to the transcriptional machinery and enhancing DNA accessibility (*Vo and Goodman, 2001*). AMPK activation enhances P300 degradation and causes phosphorylation of P300 at serine residue 89, thereby blocking the interaction of P300 with PPARγ and reducing PPARγ transcriptional activity (*Leff, 2003*; *Lim et al., 2012*; *Yang et al., 2001*). Although our *in vitro* data suggest an involvement of P300 and PPARγ in the suppression of ANGPTL4 by AMPK activation, whether P300 is also involved in *in vivo* regulation of ANGPTL4 during cold exposure remains to be determined.

Despite the unmistakable dependency of BAT LPL activity on ANGPTL4 expression, a modest cold-induced increase in LPL activity and TRL-derived fatty acid uptake is observed in BAT of *Angptl4* -/- mice, which may be (partially) explained by the moderate increase in *Lpl* mRNA in BAT in the cold. Alternatively, there may be a role for another, yet to be identified, cold-induced post-translational modulator of LPL. It may be hypothesized that this post-translational modulator is also involved in the rapid increase in LPL activity during acute cold exposure (*Klingenspor et al., 1996*).

No overt abnormalities in cold-tolerance were observed in the *Angptl4-/-* or *Angptl4*-Tg mice as compared to wild-type mice. This observation, however, does not refute the importance of ANGPTL4 in fuel delivery to BAT during cold exposure. BAT is a well-conserved organ that is postulated to have conferred to mammals the evolutionary advantage to survive cold stressors such as birth or low environmental temperatures (*Cannon and Nedergaard, 2004*). It is likely that differential uptake of TG in other organs and altered uptake of other available fuels (free fatty acids, glucose) compensate for the reduced uptake of TRL-derived fatty acids by BAT in *Angptl4*-Tg mice. Strikingly, adipose tissue-specific deletion of LPL in mice does not result in an overt phenotype (*Bartelt et al., 2012*; *Garcia-Arcos et al., 2013*). While plasma TG levels are elevated in these mice, no other parameters were altered, indicating that even in mice completely lacking LPL in adipose tissue, alternative mechanisms exist to fuel BAT and WAT (*Bartelt et al., 2012*; *Garcia-Arcos et al., 2013*).

In conclusion, our data show that regulation of ANGPTL4 is an important factor in directing lipid fuels towards BAT and away from WAT during prolonged cold exposure. Better understanding of the mechanisms underlying fuel re-distribution may pave the way for new strategies to combat metabolic diseases, such as cardiovascular disease and diabetes type 2, in which a mismatch in regulation of lipid uptake and usage by tissues is an important feature (*Klop et al., 2013*; *Young and Zechner, 2013*).

# Materials and methods

## Mice

Three- to four-month old *Angptl4-/-*, wild-type and *Angptl4*-Tg mice were either placed at a thermo-neutral temperature (~28°C) (n = 7/8, as indicated in figures) or at a cold temperature (~4°C) (n = 7–10, as indicated in figures) for a period of 10 days. All animals are backcrossed on a pure C57Bl/6J background for multiple generations (>10). Wild-type and *Angptl4*-Tg mice are littermates. *Angptl4-/-* mice have been obtained via homologous recombination of embryonic stem cells and lack part of the *Angptl4* gene, resulting in a non-functional ANGPTL4 protein (*Köster et al., 2005*; *Lichtenstein et al., 2010*). *Angptl4*-Tg mice over-express the *Angptl4* gene in various tissues under its own promoter (*Mandard et al., 2006*). Food intake, body weight and body temperature were monitored daily. Body temperature of cold-exposed mice was monitored via read-out of transponders (IPTT-300) that were injected subcutaneously prior to the experiment (Bio Medic Data Systems, Seaford, USA). The Animal Ethics Committees of Wageningen University and University Medical Center Hamburg-Eppendorf approved all experiments.

## TRL labelling

Radiolabelled VLDL-like emulsion particles were essentially prepared as described previously (*Rensen et al., 1995*). Briefly, 100 mg of lipids (triolein, egg yolk phosphatidylcholine, lysophosphatidylcholine, cholesteryl-oleate and cholesterol) were mixed with glycerol tri[$^3$H]oleate and [$^{14}$C]cholesteryl-oleate (GE Healthcare, Little Chalfont, UK) and sonicated (Soniprep 150, MSE Scientific Instruments, UK). The emulsion was fractionated by consecutive density gradient ultracentrifugation (Beckman, California, USA) to yield VLDL-like particles with a diameter of ~80 nm.

Radiolabelled chylomicron-like particles with a diameter of ~250 nm were prepared from lipids derived from human TRLs of apoCII-deficient subjects (approved by Ärtzekammer Hamburg, Germany) as described previously (*Bruns et al., 2009*). Briefly, 10 mg of isolated lipid was mixed with glycerol tri[$^{14}$C]oleate and [$^3$H]cholesteryl-oleyloleate (Perkin Elmer, Rodgau, Germany) in chloroform, after which solvent was removed and TRL particles were formed by sonication in 1 mL of PBS for 10 min at 60°C. Aggregates were removed by filtration through a 450 nm filter (Millipore). TRL particles used for intravital microscopy were prepared similarly, but radiolabels were replaced by hydrophobic fluorescent nanocrystals (QD-TRLS).

## TG clearance experiments

To study the clearance of radiolabelled TRL-like particles (80 and 250 nm), *Angptl4-/-*, wild-type and *Angptl4*-Tg mice were exposed to cold or thermo-neutral temperature for 10 days (see *Figure 4—figure supplement 1* for an overview). Prior to the experiment, animals were fasted for 4 hr. To asses TG clearance, mice were injected intravenously with 200 μL radiolabelled TRL-like particles (0.2 mg TG for VLDL-like particles, 2 mg TG for chylomicron-like particles). Lipid turnover was determined for VLDL-like particles from plasma taken at 2,5,10, and 15 min following injection. Total plasma volumes were calculated as 0.04706 x body weight (g) (*Jong et al., 2001*). 15 min after injection, mice were sacrificed and perfused via the heart with ice-cold PBS containing 50 IU/mL heparin. Multiple organs were collected, weighed and solubilized in Tissue Solubilizer (Amersham Biosciences, Roosendaal, the Netherlands; for VLDL-like particles) or Solvable (Perkin Elmer; for chylomicron-like particles) overnight. $^3$H and $^{14}$C radioactivity was determined via liquid scintillation counting. Uptake of radioactivity derived from TRL-like particles was calculated as % uptake of the injected radiolabel per gram tissue.

## Intravital microscopy

For intravital microscopy, interscapular BAT was exposed in anesthetized *Angptl4-/-*, wild-type and *Angptl4*-Tg mice and visualized by a confocal microscope with resonant scanner (Nikon A1R). QD-TRLs were injected via a tail vein catheter in anesthetized mice, followed by recording of 30 confocal images per second of the interscapular BAT were recorded for a period of 15 min. The acquired data were edited in Nikon NIS Elements. After recording, mice were perfused with PBS containing 50 IU/mL heparin. Then, BAT, gWAT and iWAT were taken for subsequent cryosectioning, to assess the uptake of QD-TRLs via confocal microscopy.

## Plasma measurements

Plasma concentrations of glucose (Sopachem, Ochten, the Netherlands), triglycerides (TG), cholesterol (Instruchemie, Delfzijl, the Netherlands), glycerol (Sigma-Aldrich, Houten, the Netherlands) and free fatty acids (Wako Chemicals, Neuss, Germany; HR(2) Kit) were determined following the manufacturers' instructions.

## LPL activity measurements

LPL activity in whole tissue homogenates from BAT, WAT and inguinal adipose tissue was measured as described previously (*Ruge et al., 2004*). Briefly, extracts of frozen tissue samples were prepared in 9 ml lysis buffer/g tissue (0.025 M NH$_3$, 5 mM Na$_2$EDTA, and per ml: 1 mg bovine serum albumin, 10 mg Triton X-100, 1 mg SDS, 5 IU heparin, and Complete protease inhibitors [Roche]) by homogenization with a Polytron homogenizer. Homogenates were spun down for 15 min at 3000 × g to obtain the supernatant used to measure LPL activity. 2 µl of supernatant was assayed, in a total volume of 200 µL, with a $^3$H-oleic acid-labelled triolein containing substrate emulsion having 100 mg soybean triglycerides and 10 mg egg yolk phospholipids per mL. Incubation was at 25°C for 30–100 min, dependent on the expected level of LPL activity. One milliunit (mU) of enzyme activity corresponds to 1 nmol of fatty acid released per min.

## RNA isolation and qPCR

Total RNA was isolated using TRIzol reagent (Life Technologies Europe BV, Bleiswijk, the Netherlands). RNA from WAT depots was purified using the Qiagen RNeasy Micro kit (Qiagen, Venlo, the Netherlands). RNA was reverse transcribed using a First-Strand cDNA Synthesis Kit (Thermo Scientific, Landsmeer, the Netherlands) (for cells) or iScript cDNA Synthesis Kit (Bio-Rad, Veenendaal, the Netherlands) (for tissues). Real-time PCR was carried out using SensiMiX (Bioline, GC Biotech, Alphen aan de Rijn, the Netherlands) on a CFX 384 Bio-Rad thermal cycler (Bio-Rad). TBP and 36B4 were used as housekeeping genes. Primer sequences can be found in *Table 1*.

## Tissue H&E staining

Fresh tissues (WAT and BAT) were fixed in 4% paraformaldehyde, dehydrated and embedded in paraffin. H&E staining was performed using standard protocols.

## Tissue immunofluorescence

Frozen human BAT sections obtained during surgery (5 µm thick) were fixated during 15 min in 3.7% formaldehyde in PBS, followed by incubation for 45 min at room temperature with a primary antibody (polyclonal rabbit hANGPTL4 or a polyclonal rabbit UCP1-antibody [kind gift of Dr. B. Cannon, Stockholm University]) (*Vijgen et al., 2013*; *Wu et al., 2012*) diluted in 0.05% Tween20 in PBS. After three washing steps with PBS, sections were incubated for 45 min at room temperature with the appropriate fluorescently labelled secondary antibodies. The specificity of the antibody for ANGPTL4 was demonstrated previously via immunoblot of human plasma using appropriate peptide

**Table 1.** Primer sequences.

| Gene | Forward primer | Reverse primer |
| --- | --- | --- |
| m*36b4* | ATGGGTACAAGCGCGTCCTG | GCCTTGACCTTTTCAGTAAG |
| m*Angptl4* | GTTTGCAGACTCAGCTCAAGG | CCAAGAGGTCTATCTGGCTCTG |
| m*Lpl* | GGGAGTTTGGCTCCAGAGTTT | GGGAGTTTGGCTCCAGAGTTT |
| m*Ucp1* | CCTGCCTCTCTCGGAAACAA | TGTAGGCTGCCCAATGAACA |
| m*Pgc1α* | AGTCCCATACACAACCGCAGTCGCAACATG | CCCTTTCTTGGTGGAGTGGCTGCCTTGG |
| m*Cidea* | TGACATTCATGGGATTGCAGAC | GGCCAGTTGTGATGACTAAGAC |
| m*Elovl3* | TTCTCACGCGGGTTAAAAATGG | GAGCAACAGATAGACGACCAC |
| m*Prdm16* | CCACCAGCGAGGACTTCAC | GGAGGACTCTCGTAGCTCGAA |

controls and was validated by staining of ANGPTL4 in human heart, intestine and muscle (*Alex et al., 2014*; *Catoire et al., 2014*; *Georgiadi et al., 2010*).

## Human cold exposure experiment and hANGPTL4 ELISA

Plasma ANGPTL4 levels were measured in plasma samples from a published study in which lean and obese human subjects were exposed to mild cold (16°C) for 48 hr (*Wijers et al., 2010*). Plasma hANGPTL4 levels were measured as described previously (*Kersten et al., 2009*). Briefly, 96-well plates were coated with anti-human ANGPTL4 polyclonal goat IgG antibody (AF3485; R&D Systems) and were incubated overnight at 4°C. After blocking, 100 µL of 20-fold diluted human plasma was applied to each well, followed by incubation at room temperature for 2 hr. Next, 100 µL of diluted biotinylated anti-human ANGPTL4 polyclonal goat IgG antibody (BAF3485; R&D Systems) was added and incubated for 2 hr. Subsequently, streptavidin-conjugated HRP was added for 20 min, followed by tetramethylbenzidine substrate reagent for 6 min. The reaction was stopped by adding 50 µL of 10% $H_2SO_4$. The absorbance was measured at 450 nm.

## Western blot

Tissues were lysed in a mild RIPA-like lysis buffer (25 mM Tris-HCl pH 7.4, 150 mM NaCl, 1 mM EDTA, 1% NP-40 and 5% glycerol; Thermo Scientific) with protease and phosphatase inhibitors (Roche). Cells were lysed directly in 2x Laemmli sample buffer (LSB) with DTT. Protein lysates (20–30 µg protein per lane) were loaded on a denaturing gel (Bio-Rad) and separated by SDS gel electrophoresis. Proteins were transferred to a PVDF membrane by means of a Transblot Turbo System (Bio-Rad). The primary antibody [rabbit anti–phospho-AMPK antibody (Thr172), rabbit anti-AMPKα1,α2 antibody (Cell Signaling Technology, #2535 and #2532), rabbit anti-mouse/human AMPKα1 (Cell Signaling Technology, #2795), rabbit anti-mouse/human AMPKβ1 and AMPKβ2 (Cell Signaling Technology, #4148 and #4178), rabbit anti-mouse AMPKα2 antibody (Abcam, #ab3760), rat anti-mouse ANGPTL4 antibody (Adipogen, #Kairos 142–2), goat anti-mouse LPL antibody (kind gift from André Bensadoun), rabbit anti-mouse HSP90 antibody (Cell Signaling Technology, #4874S), or mouse anti-mouse β-tubulin antibody (Santa-Cruz Biotechnology, #sc23949)] was used at a ratio of 1:1000 (AMPKα1,α2, AMPKα2, AMPKα1, AMPKβ1, AMPKβ2, ANGPTL4, HSP90, β-tubulin), 1:2000 (phospho–AMPK) or 1:5000 (LPL). Corresponding secondary antibodies (HRP-conjugated) (Sigma-Aldrich) were used at 1:5000 dilutions. All incubations were performed in Tris-buffered saline, pH 7.5, with 0.1% Tween-20 (TBS-T) and 5% dry milk, except for anti-AMPKα1,2 and anti-phospho-AMPK Thr172 antibody, where 5% bovine serum albumin (BSA) was used instead of milk. All washing steps were in TBS-T without dry milk or BSA. Blots were visualized using the ChemiDoc MP system (Bio-Rad) and Clarity ECL substrate (Bio-Rad).

## Cell culture

3T3-F442a cells (P8-P14, Sigma) were maintained in DMEM (Lonza), supplemented with 10% newborn calf serum and 1% penicillin/streptomycin (P/S) under 5% $CO_2$ at 37°C. At confluency, cells were switched to DMEM (Lonza, Verviers, Belgium), supplemented with 10% fetal bovine serum (FBS), 1% P/S and 5 µg/mL insulin (Sigma-Aldrich) to stimulate differentiation. During differentiation, medium was changed every 2-3 days. After 10 days of differentiation, cells were switched back to regular medium for 2-3 days, after which experiments were performed.

T37i cells (P31-36; kind gift of Marc Lombès) were cultured in DMEM/F-12 (Gibco, Life Technologies, Blijswijk, the Netherlands), supplemented with 10% FBS and 1% P/S. Two days post-confluency, cell culture medium was supplemented with 112 ng/mL insulin and 2 nM T3 (Sigma-Aldrich) to induce differentiation. After 7 days of differentiation, cells were switched back to regular medium and used for experiments 2-3 days after (*Zennaro et al., 1998*).

HepG2 cells (passage unknown) were maintained in DMEM (Lonza) supplemented with 10% FBS and 1% P/S. At each passage, the cell pellet was filtered through a 40 µm filter to reduce cell clumping.

Culture of BA adipocytes was performed as described previously (*Uldry et al., 2006*). Briefly, immortalized brown adipocytes are grown to confluence with differentiation medium (DMEM, 10% FBS, 20 nM insulin, 1 nM T3). Upon confluence, cells were treated with induction medium

(differentiation medium supplemented with 0.5 mM IBMX, 0.5 µM dexamethasone, 0.125 mM indomethacin) for two days. After washing, cells were incubated in differentiation medium for another 5 to 7 days.

## Isolation of primary adipocytes

WAT was dissected from C57Bl/6J mice and put in DMEM supplemented with 1% P/S and 1% BSA. Tissues of 3–4 mice were pooled, minced with scissors and digested for 1 hr at 37°C in collagenase-containing medium (DMEM with 3.2 mM CaCl$_2$, 1.5 mg/mL Collagenase type II (Sigma-Aldrich), 10% FBS, 0.5% Bovine Serum Albumin (Sigma-Aldrich) and 15 mM HEPES, filtered). After digestion, cell mixture was passed over a 100 µm cell strainer and centrifuged at 1600 rpm for 10 min. Supernatant was removed and the pellet containing the stromal vascular fraction was re-suspended in erythrocyte lysis buffer (155 mM NH$_4$Cl, 12 mM NaHCO$_3$, 0.1 mM EDTA) and incubated for 2–3 min at room temperature. Following neutralization, cells were centrifuged at 1200 rpm for 5 min. Cells were re-suspended in DMEM containing 10% FBS and 1% P/S and plated. Upon confluence, cells were differentiated following standard protocol of 3T3-L1 cells, as described previously (*Alex et al., 2013*).

For isolation of primary brown adipocytes, BAT from 1-month-old pups was used. Tissues of 5–10 pups were pooled, minced with scissors and digested for 30 min in collagenase-containing medium at 37°C (DMEM w/o serum, 2 mg/mL Collagenase type II, 2% BSA, 25 mM HEPES). After digestion, cells were passed through a 70 or 100 µM filter, mature adipocytes were discarded and cells were centrifuged at 800*g for 5 min. Cells were re-suspended in differentiation medium (DMEM, 10% FBS, 20 nM insulin, 1 nM T3) and plated. Upon confluence, cells were treated with induction medium (differentiation medium supplemented with 0.5 mM IBMX, 0.5 µM dexamethasone, 0.125 mM indomethacin) for two days. After washing, cells were incubated in differentiation medium for another 5 to 7 days.

## siRNA experiments

T37i cells were cultured and differentiated as described above. At day 10 of differentiation, mature adipocytes were trypsinized and replated at 70% density. 2 hr post-plating, siRNAs against AMPKα1 and AMPKα2, or non-target control (Dharmacon, via Thermo Fisher) complexed to Lipofectamine RNAimax reagent (Life Technologies) were added, according to the manufacturer's protocol. 48 hr post-transfection, cells were washed and treated as indicated before harvesting for RNA analyses.

## BAT and WAT explants

BAT and WAT tissues were dissected from male C57Bl/6J mice and transferred to DMEM supplemented with 1% P/S and 1% BSA. Tissues of 4 mice were pooled and minced finely with scissors. Approximately 50 mg of tissue was transferred to a well of a 24-wells tissue culture plate and equilibrated for 1 hr in DMEM, supplemented with 1% P/S. After 1 hr, explants were treated as indicated before being harvested for RNA and protein analyses.

## Chemicals

Isoproterenol, AICAR, metformin, rosiglitazone, Wy14,643, GW510516, GW742, dexamethasone, insulin, 3-isobutyl-1-1methylxanthine (IBMX), actinomycin D were purchased from Sigma-Aldrich. A769662 was purchased from Abcam (Cambridge, United Kingdom), phenformin hydrochloride was purchased from Cayman Chemicals (via SanBio, Uden, the Netherlands).

## mAngptl4 PPRE construct

A fragment of 1517 bp containing intron 3 of the mouse *Angptl4* gene was amplified from DNA of the mouse *Angptl4* gene using the primers: Fwd 5'-TTGCTGTCATCTGGCAACTC-3' and Rev 3'-CACCTAAAGCCTACCCCACA-5'. The resulting fragment was gel-purified using QIAquick Gel Extraction Kit (Qiagen) and subjected to a second PCR to specifically amplify the three functional PPREs of *Angptl4* and to introduce *Xho1* and *Kpn1* restriction sites. Amplification of the 565 bp fragment was done with the following primers: mouse Fwd 5'-atggtaccTTCACACCCTAAGGCTGC-3', and mouse Rev 3'-atctcgagGGGGGAAGAGGAAGAAAA-5'. Following purification from gel, the fragment was subjected to restriction with *Xho1* and *Kpn1* enzymes and cloned into the *Xho1* and *Kpn1* sites of the pGL3 promoter vector (Promega, Leiden, the Netherlands). Presence of the correct insert was

validated by sequencing (EZ-Seq, Macrogen, Amsterdam, the Netherlands) using a RV3 primer: '5-C-TAGCAAAATAGGCTGTCCC-3'.

## Transactivation assays

m*Angptl4* PPRE pGL3 reporter vector was transfected into the human hepatocellular cell line HepG2 (ATCC, Manassas, USA) in the presence or absence of pSG5 vectors expressing *Ppar*γ and *Rxr* and pcDNA vector expressing HA-*P300* (kind gift of Eric Kalkhoven, University Medical Centre Utrecht, the Netherlands). A vector expressing renilla luciferase under a SV40 promoter was co-transfected with all samples to determine transfection efficiency. Transfections were performed using polyethylenimine (PEI) (Polysciences Inc. via Qiagen). 16 hr post-transfection cells were incubated with rosiglitazone (5 µM) and/or AICAR (1 mM) for 9 hr. Firefly and renilla luciferase activity were determined using the Dual Glo Luciferase Assay system (Promega), according to manufacturer's instructions, on a Fluroskan Ascent apparatus (Thermo Scientific).

## Statistics

Data are expressed as mean ± SEM, unless otherwise indicated. Differences were evaluated for statistical significance by student t-test or two-way ANOVA, followed by a post-hoc Tukey HSD test, and considered statistically significant when $p < 0.05$.

## Acknowledgements

We would like to thank Desirée Veening-Griffioen, Georgia Lenihan-Geels, Frits Mattijssen, Solveig Nilsson, Sander Kooijman, Padmini Khedoe, Geerte Hoeke, Jimmy Berbée, Andrea van Dam, Martina Jakubikova, Sandra Ehret, Ludger Scheja and Rinke Stienstra for excellent assistance. We would also like to thank Sander Wijers for the plasma samples of lean and obese individuals exposed to mild cold. We are grateful for the kind gift of T37i cells by Marc Lombès. This study was supported by grant 12CVD04 from the Fondation Leducq and by grant 12203 from the Swedish Research Council. PCN Rensen is Established Investigator of the Dutch Heart Foundation (2009T038).

## Additional information

### Funding

| Funder | Grant reference number | Author |
| --- | --- | --- |
| Fondation Leducq | 12CVD04 | Wieneke Dijk<br>Markus Heine<br>Laurent Vergnes<br>Karen Reue<br>Joerg Heeren<br>Sander Kersten |
| Forskningsrådet om Hälsa, Arbetsliv och Välfärd | 12203 | Gunilla Olivecrona |
| Hartstichting | 2009T038 | Patrick CN Rensen |

The funders had no role in study design, data collection and interpretation, or the decision to submit the work for publication.

### Author contributions

WD, MRB, SK, Conception and design, Acquisition of data, Analysis and interpretation of data, Drafting or revising the article; MH, Conception and design, Acquisition of data, Analysis and interpretation of data; LV, GO, Acquisition of data, Analysis and interpretation of data, Drafting or revising the article; GS, Acquisition of data, Analysis and interpretation of data; MKCH, Analysis and interpretation of data, Drafting or revising the article; KR, Conception and design, Drafting or revising the article; WDvanML, Conception and design, Analysis and interpretation of data; PCNR, JH, Conception and design, Analysis and interpretation of data, Drafting or revising the article

## Ethics

Human subjects: All subjects signed an informed consent for the study protocol, which was approved by the institutional review board of Maastricht University Medical Centre.

Animal experimentation: This study was performed in accordance with Directive 2010/63/EU from the European Union. All of the animals were handled according to protocols approved by the Animal Ethics Committees of Wageningen University and Hamburg University (2013007.d, 2013100.b and 34/12).

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
