## [Decision Letter]

[Editors’ note: this article was originally rejected after discussions between the reviewers, but the authors were invited to resubmit after an appeal against the decision.]

Thank you for choosing to send your work entitled "ANGPTL4 mediates shuttling of lipid fuel to brown adipose tissue during sustained cold exposure" for consideration at *eLife*. Your full submission has been evaluated by Mark McCarthy (Senior Editor) and two peer reviewers, one of whom is a member of our Board of Reviewing Editors, and the decision was reached after discussions between the reviewers. Based on our discussions and the individual reviews below, we regret to inform you that your work will not be considered further for publication in *eLife*.

The reason for the negative decision is that very substantial additional experiments would be necessary to solidify the direct involvement of AMPK in the mechanism of ANGPTL4 regulation. As it stands, this connection is weak and not substantiated with sufficient evidence. The literature on AMPK reveals the difficulty in relying on AICAR to document direct involvement on AMPK in biological pathways. There are other problematical issues such as the apparent inconsistencies in Figure 5 that contradict the title of that section of the paper. Although the manuscript presents an interesting story, the mechanistic information remains limited at this stage of the study. As you know, *eLife*'s policy is not to make offers of resubmission that would be contingent on the generation of extensive additional data: rather it makes a judgement on the paper "as is", allowing relatively rapid resubmission elsewhere if the paper is not taken forward.

Reviewer #1:

Djik et al. present compelling evidence that ANGPTL4 mediates trafficking of lipids to brown adipose during cold exposure. The experiments outlined, and the models chosen to analyze convincingly demonstrate that alterations in the local levels of ANGPTL4 modulate LPL activity differentially between white and brown adipose depots. This work is performed well, is concise, and marks an interesting and unique finding explaining a previously undescribed physiological phenomenon. A few points that deserve further attention:

1) The presumed physiological rationale for increased lipid delivery to BAT during cold exposure is for increased thermogenesis. However, the *Angptl4* transgenic mice have no alteration in core body temperature following cold exposure (Figure 2). How do the authors rationalize this observation? And further, does decreased LPL activity in BAT and WAT during cold exposure in the *Angptl4*-TG alter weight balance upon cold exposure?

2) While Figure 5 nicely demonstrates that *Angptl4* expression may limit increases in LPL activity in WAT following cold exposure, Figure 5 directly contradicts the title to this section (*Angptl4* expression and LPL activity are oppositely regulated in WAT and BAT upon sustained cold exposure) as 5I and J clearly indicate increased LPL activity in IWAT following cold exposure. This is in stark contrast to Figure 5 and the message of the paper which indicates increased ANGPTL4 causes decreased lipid uptake. Finally, nowhere in this figure are the protein levels of ANGPTL4 demonstrated to be altered in WAT following cold exposure.

While this does not alter the entire findings of the paper, and the following statement remains true – "*Angptl4* expression caused a clear dose-dependent reduction in uptake of TRL-derived fatty acids" –, the authors must reconcile these findings, potentially by performing these experiments after a shorter cold exposure time before significant browning has taken place in the Inguinal WAT depot.

3) The authors argue that "the opposite regulation of ANGPTL4 expression in BAT and WAT may be mediated by differential activation of AMPK"). However Figure 8 shows that cold exposure increases P-AMPK in both iWAT and BAT despite that these tissues display differential regulation of ANGPL4. Although the other experiments here support their claim as AiCAR treatment decreases ANGTPL4 expression, the purported mechanism of differential regulation via AMPK seems unlikely and further work is needed to elaborate this claim.

4) ANGTPL4 protein is processed and can be released into circulation as both a paracrine and an endocrine factor. This has been shown to have important clinical implications (Arteriosclerosis, Thrombosis, and Vascular Biology. 2009; 29: 969-974). How are the levels of this protein altered during cold exposure, and do these alterations follow the authors' reported findings?

5) Expand on LPL activity assay. This is a commonly misunderstood assay and it is imperative the reader be provided an accurate method of analysis.

Reviewer #2:

The issue addressed by this paper – how is fuel provided to BAT during cold exposure – is a critical question in adipose biology and of high interest in the field. The authors provide strong additional evidence that modulating LPL activity is a principal route by which lipid substrate is directed into BAT in the cold, consistent with previous work by this group and other laboratories. The concept that *Angptl4* downregulation by cold exposure, specifically in BAT, is a key process in this pathway appears solid and does provide new information to the field.

A major issue with the study at its present stage is the relatively weak data on mechanism, particularly with regards to the potential role of AMPK in this system. Definitive data showing AMPK mediates the effect of cold on *Angptl4* transcription would bring this study up to the level of high significance. The use of AICAR is equivocal, as other effects of AICAR have been shown to be mediated by pathways independent of AMPK and it is not a fully definitive reagent to use to solidify a conclusion about AMPK. While the data with AICAR are instructive and suggestive, the authors need to provide genetic or other evidence that AMPK is directly mediating the effect of cold exposure on ANGPTL4 expression. Providing this one additional key piece of data would greatly enhance the impact of the paper.

[Editors’ note: what now follows is the decision letter after the authors submitted for further consideration.]

Thank you for resubmitting your work entitled "ANGPTL4 mediates shuttling of lipid fuel to brown adipose tissue during sustained cold exposure" for further consideration at *eLife*. Your revised article has been favorably evaluated by Mark McCarthy (Senior Editor) and a Reviewing Editor. The manuscript has been accepted in principle, but there are some remaining issues that need to be addressed before publication, as outlined below:

Please include the data on iWAT (which was taken out of the original manuscript) and data on the human tissue (which you included in the rebuttal but not the manuscript) in the supplemental section, with reference to those data in the paper.

---

## [Author Response]

[Editors’ note: the author responses to the first round of peer review follow.]

Given the overall very positive response of the reviewers to our manuscript, we were surprised by the decision to reject it. The main stumbling block appeared to be the insufficient evidence for a direct involvement of AMPK in the mechanism of ANGPTL4 and LPL regulation during cold in white and brown adipose tissue. This issue was raised by both reviewers but no specific suggestions for experiments were raised, as both reviewers must have recognized the near impossibility of addressing that question via in vivo experiments, short of investing at least 2-3 years of intense work. Consequently, we further elaborated on the connection between AMPK and ANGPTL4 via in vitro experiments. We feel that in the revised manuscript, containing more extensive in vitro data confirming the involvement of AMPK in ANGPTL4 regulation, the issues raised by the reviewers have been largely resolved. The following major additions and changes were made to the paper:

1) We added data on WAT and BAT weights as well as bodyweight gain during cold exposure;

2) We have added immunoblots of various AMPK subunits in white and brown adipose tissue. The marked difference in abundance of various AMPK subunits between brown and white adipose tissue provides further insight into differences in AMPK activation and expression between the two tissues;

3) We show that a down-regulation of *Angptl4* mRNA expression in brown adipocytes is not exclusive to AICAR, but is also elicited by activation of AMPK by metformin, A-769662 and phenformin hydrochloride;

4) We confirm the role of AMPK in the negative regulation of *Angptl4* by treatment of brown adipocytes with siRNA against AMPKα1 and AMPKα2. The down-regulation of *Angptl4* upon treatment with A-769662 and AICAR was greatly attenuated in siAMPK-treated adipocytes;

5) We have added data showing that AMPK activation of BAT explants but not WAT explants causes down-regulation of *Angptl4*, highlighting the different responses of the two tissues upon AMPK activation;

6) We mostly removed the data on inguinal white adipose tissue from the manuscript. Judging from the reviewers’ comments, the addition of inguinal white adipose tissue, a tissue with properties in between gonadal white adipose tissue and brown adipose tissue, was confusing rather than complementary and supportive for our mechanism;

7) We leave open the possibility of adding new data showing an increase in plasma ANGPTL4 in obese individuals subjected to two days of mild cold (see response to last comment of reviewer 1);

8) Finally, as requested, we have further expanded the protocol of the LPL activity measurements. Reviewer #1:*Djik et al. present compelling evidence that ANGPTL4 mediates trafficking of lipids to brown adipose during cold exposure. The experiments outlined, and the models chosen to analyze convincingly demonstrate that alterations in the local levels of ANGPTL4 modulate LPL activity differentially between white and brown adipose depots. This work is performed well, is concise, and marks an interesting and unique finding explaining a previously undescribed physiological phenomenon. A few points that deserve further attention:*

*1) The presumed physiological rationale for increased lipid delivery to BAT during cold exposure is for increased thermogenesis. However, the* Angptl4 *transgenic mice have no alteration in core body temperature following cold exposure (Figure 2). How do the authors rationalize this observation? And further, does decreased LPL activity in BAT and WAT during cold exposure in the* Angptl4*-Tg alter weight balance upon cold exposure?*

Given the importance of cold-induced thermogenesis in mice, there is substantial redundancy in fuel supply to BAT. It is likely that *Angptl4*-Tg mice compensate for the lower lipid uptake by increased uptake of another fuel, e.g. glucose. It should be emphasized that very few mutant mice show a reproducible defect in cold-tolerance, including PPARa knock-out mice and adipose tissue-specific LPL knock-out mice. Compensatory mechanisms may also explain why after cold exposure we do not observe a difference in weight in *Angptl4*-Tg mice as compared to *Angptl4*^-/-^ and *Angptl4 ^/^* mice.

Consistent with what would be expected, under thermoneutral conditions weight of WAT and BAT were highest in the *Angptl4*-Ko mice and lowest in the *Angptl4*-Tg mice. In all genotypes it was observed that weight of WAT was lower in cold-exposed mice as compared to mice kept at thermoneutral temperatures. In contrast, in all genotypes weight of BAT was higher in cold-exposed mice as compared to mice kept at thermoneutral temperatures. For reasons that are unclear, the relative difference in weight of WAT between cold-exposed and thermoneutral mice was lowest in the *Angptl4*-Tg group. The data have been added to Figure 2. Finally, in agreement with the lack of difference in food intake between the three genotypes, bodyweight gain during the 10 day cold exposure did not differ between the three genotypes. The data have been added to Figure 2.

*2) While Figure 5 nicely demonstrates that* Angptl4 *expression may limit increases in LPL activity in WAT following cold exposure, Figure 5 directly contradicts the title to this section (*Angptl4 *expression and LPL activity are oppositely regulated in WAT and BAT upon sustained cold exposure) as 5I and J clearly indicate increased LPL activity in IWAT following cold exposure. This is in stark contrast to Figure 5 and the message of the paper which indicates increased ANGPTL4 causes decreased lipid uptake. Finally, nowhere in this figure are the protein levels of ANGPTL4 demonstrated to be altered in WAT following cold exposure*.

*While this does not alter the entire findings of the paper, and the following statement remains true – "*Angptl4 *expression caused a clear dose-dependent reduction in uptake of TRL-derived fatty acids" –, the authors must reconcile these findings, potentially by performing these experiments after a shorter cold exposure time before significant browning has taken place in the Inguinal WAT depot.*

We would like to emphasize that inguinal WAT is an atypical fat depot that displays substantial browning, in contrast to regular (gonadal) WAT. As a result, the response of iWAT to cold contains features of both BAT and regular WAT, consistent with the mixed cell population within iWAT following cold. We had felt that inclusion of iWAT in the paper would strengthen our mechanism, showing a continuum between different tissues with different amounts of ‘brown-like’ cells. However, we recognize that it may also create confusion by not presenting a uniform phenotype of WAT, as suggested by the section title and the Abstract. Based on this consideration, we decided to remove the data on iWAT to present a more black and white story comparing BAT with regular (gonadal) WAT. Protein levels of ANGPTL4 in BAT and WAT upon cold exposure can be found in Figure 5, showing a clear decrease in ANGPTL4 protein in BAT and a clear decrease in ANGPTL4 protein in WAT by cold.

*3) The authors argue that "the opposite regulation of ANGPTL4 expression in BAT and WAT may be mediated by differential activation of AMPK". However Figure 8 shows that cold exposure increases P-AMPK in both iWAT and BAT despite that these tissues display differential regulation of ANGPL4. Although the other experiments here support their claim as AiCAR treatment decreases ANGTPL4 expression, the purported mechanism of differential regulation via AMPK seems unlikely and further work is needed to elaborate this claim*.

We agree that further elaboration on the mechanism would strengthen the manuscript. Consequently, in the revised manuscript several additions were made to specifically elaborate on the opposite regulation of ANGPTL4 in BAT and regular (gonadal) WAT is regulated via AMPK: 1) We have added immunoblots of various AMPK subunits in white and brown adipose tissue. The marked difference in abundance of various AMPK subunits between brown and white adipose tissue provides further insight into differences in AMPK activation and expression between the two tissues; 2) We show that a down-regulation of *Angptl4* mRNA expression in brown adipocytes is not exclusive to AICAR, but is also elicited by activation of AMPK by metformin, A-769662 and phenformin hydrochloride; 3) We confirm the role of AMPK in the negative regulation of *Angptl4* by treatment of brown adipocytes with siRNA against AMPKα1 and AMPKα2. The down-regulation of *Angptl4* upon treatment with A-769662 and AICAR was greatly attenuated in siAMPK-treated adipocytes; 4) We have added data showing that AMPK activation of BAT explants but not WAT explants causes down-regulation of *Angptl4*, highlighting the different responses of the two tissues upon AMPK activation.

With respect to the comment on p-AMPK in iWAT and BAT, we would like to emphasize that while P-AMPK is indeed slightly higher in iWAT upon cold exposure, this is not the case for gWAT (was Figure 8—figure supplement 1, now Figure 7), which shows no change in p-AMPK. Accordingly, we believe that the increase of p-AMPK in iWAT is due to browning. As discussed above, iWAT exhibits a phenotype intermediate between gWAT and BAT. As suggested in our response to the second comment, we decided to remove iWAT from the paper to prevent any misunderstandings.

*4) ANGTPL4protein is processed and can be released into circulation as both a paracrine and an endocrine factor. This has been shown to have important clinical implications (Arteriosclerosis, Thrombosis, and Vascular Biology. 2009; 29: 969-974). How are the levels of this protein altered during cold exposure, and do these alterations follow the authors' reported findings?*

We thank the reviewer for this insightful comment. We have tested all commercial ELISAs for mouse ANGPTL4 but unfortunately none allows detection of ANGPTL4 protein in mouse plasma. With respect to humans, we have measured plasma ANGPTL4 in human subjects exposed to 16ºC for 48 hours (Wijers et al., Obesity (Silver Spring) 2010). Interestingly, we found that mild cold caused a statistically significant increase in plasma ANGPTL4 in obese subjects but not in lean subjects. One possibility for the increase in plasma ANGPTL4 in obese but not in lean individuals is that obese individuals have more WAT and that the change in ANGPTL4 production in WAT dominates the amount of ANGPTL4 in plasma.

However, since ANGPTL4 is expressed in many tissues and the origin of circulating ANGPTL4 is unclear, we are a bit uncomfortable using these data in our manuscript. For this reason, we did not add a figure to the manuscript, but we are willing to add it to our paper if the reviewers and editors would consider it worthwhile.

*5) Expand on LPL activity assay. This is a commonly misunderstood assay and it is imperative the reader be provided an accurate method of analysis.*

We have further expanded the description of the LPL activity assay.

Reviewer #2*:The issue addressed by this paper* – *how is fuel provided to BAT during cold exposure* – *is a critical question in adipose biology and of high interest in the field. The authors provide strong additional evidence that modulating LPL activity is a principal route by which lipid substrate is directed into BAT in the cold, consistent with previous work by this group and other laboratories. The concept that* Angptl4 *downregulation by cold exposure, specifically in BAT, is a key process in this pathway appears solid and does provide new information to the field. A major issue with the study at its present stage is the relatively weak data on mechanism, particularly with regards to the potential role of AMPK in this system. Definitive data showing AMPK mediates the effect of cold on* Angptl4 *transcription would bring this study up to the level of high significance. The use of AICAR is equivocal, as other effects of AICAR have been shown to be mediated by pathways independent of AMPK and it is not a fully definitive reagent to use to solidify a conclusion about AMPK. While the data with AICAR are instructive and suggestive, the authors need to provide genetic or other evidence that AMPK is directly mediating the effect of cold exposure on ANGPTL4 expression. Providing this one additional key piece of data would greatly enhance the impact of the paper.*

Although we recognize the appeal of demonstrating a direct in vivo role of AMPK in mediating the effect of cold exposure on ANGPTL4 expression, it must be realized that numerous confounding factors prevent us from addressing that question via in vivo experiments (although we are open to specific suggestions). One confounding factor is that the effects of AMPK on fuel use during cold are likely also mediated by factors other than ANGPTL4. Another confounding factor is that modulating AMPK in vivo is expected to alter plasma levels of FFA, which are very potent activators of ANGPTL4 production. The only way to study the direct impact of AMPK on ANGPTL4 is via additional in vitro experiments, as described in the revised manuscript. Specifically, the following new experiments were performed and added to the manuscript.

1) We have added immunoblots of various AMPK subunits in white and brown adipose tissue. The marked difference in abundance of various AMPK subunits between brown and white adipose tissue provides further insight into differences in AMPK activation and expression between the two tissues. 2) We show that a down-regulation of *Angptl4* mRNA expression in brown adipocytes is not exclusive to AICAR, but is also elicited by activation of AMPK by metformin, A-769662 and phenformin hydrochloride. 3) We confirm the role of AMPK in the negative regulation of *Angptl4* by treatment of brown adipocytes with siRNA against AMPKα1 and AMPKα2. The down-regulation of *Angptl4* upon treatment with A-769662 and AICAR was greatly attenuated in siAMPK-treated adipocytes. 4) We have added data showing that AMPK activation of BAT explants but not WAT explants causes down-regulation of Angptl4, highlighting the different responses of the two tissues upon AMPK activation.

[Editors’ note: the author responses to the re-review follow.]

*Thank you for resubmitting your work entitled "ANGPTL4 mediates shuttling of lipid fuel to brown adipose tissue during sustained cold exposure" for further consideration at* eLife. *Your revised article has been favorably evaluated by Mark McCarthy (Senior Editor) and a Reviewing Editor. The manuscript has been accepted in principle, but there are some remaining issues that need to be addressed before publication, as outlined below: Please include the data on iWAT (which was taken out of the original manuscript) and data on the human tissue (which you included in the rebuttal but not the manuscript) in the supplemental section, with reference to those data in the paper.*

We have added data on plasma ANGPTL4 levels in lean and obese individuals exposed to a mild cold (16˚C) for 48h (Figure 5—figure supplement 1). Dr. Wouter van Marken Lichtenbelt, who provided us with these samples, has been included as a co-author on the manuscript. Descriptions of the human study, as well as of the method to measure plasma ANGPTL4 levels in human plasma have been added to the Materials and methods section.

We have re-included the data on LPL activity, Angptl4 mRNA and Lpl mRNA (Figure 5—figure supplement 2) in inguinal white adipose tissue. Data on TRL clearance by inguinal adipose tissue are presented in Figure 6. Reference to these data is made in the text when describing Figure 5.

We have added three additional headers to clarify the different sections of the manuscript.